# Mimicking To Dominate: Imitation Learning Strategies for Success in Multiagent Games

**The Viet Bui**
Singapore Management University, Singapore
`theviet.bui.2023@phdcs.smu.edu.sg`

**Tien Mai**
Singapore Management University, Singapore
`atmai@smu.edu.sg`

**Thanh Hong Nguyen**
University of Oregon Eugene, Oregon, United States
`thanhhng@cs.uoregon.edu`

## Abstract

Training agents in multi-agent games presents significant challenges due to their intricate nature. These challenges are exacerbated by dynamics influenced not only by the environment but also by strategies of opponents. Existing methods often struggle with slow convergence and instability. To address these challenges, we harness the potential of imitation learning (IL) to comprehend and anticipate actions of the opponents, aiming to mitigate uncertainties with respect to the game dynamics. Our key contributions include: (i) a new multi-agent IL model for predicting next moves of the opponents — our model works with hidden actions of opponents and local observations; (ii) a new multi-agent reinforcement learning (MARL) algorithm that combines our IL model and policy training into one single training process; and (iii) extensive experiments in three challenging game environments, including an advanced version of the Star-Craft multi-agent challenge (i.e., SMACv2). Experimental results show that our approach achieves superior performance compared to state-of-the-art MARL algorithms.

## 1 Introduction

Recent works in MARL have made a significant progress in developing new effective algorithms that can perform well in complex multi-agent environments including SMAC [34, 25]. Among these works, centralized training and decentralized execution (CTDE) [5] has attracted a great attention from the RL community due to its advantage of leveraging global information to train a centralized critic (i.e., actor-critic methods [18]) or a joint Q-function (i.e., value-decomposition methods [23, 29]). This approach enables a more efficient and stable learning process while allowing agents to act in a decentralized manner. Under this CTDE framework, off-policy methods such as MADDPG [18] and QMIX [23] have become very popular due to their data efficiency and state-of-the-art (SOTA) results on a wide range of benchmarks. On the other hand, on-policy gradient methods have been under-explored in MARL due to their data consuming and difficulty in transferring knowledge from single-agent to multi-agent settings. However, a recent work shows that on-policy methods (such as MAPPO, a multi-agent version of proximal policy optimization) outperforms all other SOTA methods including MADDPG and QMIX in various multi-agent benchmarks, and especially works

38th Conference on Neural Information Processing Systems (NeurIPS 2024).

well in complex SMAC settings [34]. Motivated by this promising result, we focus on improving the performance of policy gradient methods in MARL.

We consider a partially observable MDP environment in which there are agents attempting to form an alliance to play against a team of opponents, where allied agents have to make decision independently without communicating with other members. We aim to enhance the performance of PPO in MARL with the introduction of a novel opponent-imitation component. This new component is then integrated into the MAPPO framework to enhance the policy learning of allied agents. A key challenge in our problem setting is that allied agents are unaware of actions taken by their opponents. In addition, each allied agent only has local observations of opponents locating in the current neighborhood of the agent — the locations and neighborhoods of all players are changing over time depending on actions taken by players and the dynamics of the environment. Lastly, learning to imitate opponents occurs during the policy learning process of the allied agents. The inter-dependency between these two learning components makes the entire learning process significantly challenging.

We address these challenges while providing the following key contributions. *First*, we convert the problem of imitating the opponent policy into predicting their next states. The outcome of this next state prediction is an indirect implication of the opponent policy. We then cast the problem of opponent next-state prediction as a new multi-agent imitation learning (IL) problem. We propose a new multi-agent IL algorithm, which is an adaptation of IQ-Learn [9] (a SOTA IL algorithm), that only considers local opponent-state-only observations. Especially, instead of imitating the opponents' policy, our IL algorithm targets the prediction of next states of the neighboring opponents. *Second*, we provide a comprehensive theoretical analysis which provides bounds on the impact of the changing policy of the allied agents (as a result of the policy learning process) on our IL outcomes.

*Third*, we present a unified MARL algorithmic framework in which we incorporate our IL component into MAPPO. Our idea is to combine each allied agent's local observations with the next-state prediction of neighboring opponents of that agent, creating an augmented input based on which to improve the decision making of the allied agent at every state. This novel integration results in a new MARL algorithm, which we name *Imitation-enhanced Multi-Agent EXtended PPO* (IMAX-PPO).

*Finally*, we conduct extensive experiments in several benchmarks ranging from complex to simple ones, including: SMACv2 (an advanced version of the Star-Craft multi-agent challenge) [4], Google research football (GRF) [15], and Gold Miner [7]. Our empirical results show that our new algorithm consistently outperforms SOTA algorithms significantly accross all these benchmarks.

## 2   Related Work

**MARL.**  The literature on MARL includes both centralized and decentralized algorithms. While centralized algorithms [2] learn a single joint policy to produce joint actions of all the agents, decentralized learning [17] optimizes each agent's local policy independently. There are also algorithms based on *centralized training and decentralized execution* (CTDE). For example, methods in [18, 5] adopt actor-critic structures and learn a centralized critic that takes global information as input. Value-decomposition (VD) is a class of methods that represent the joint Q-function as a function of agents' local Q-functions [29, 23]. Alternatively, the use of policy-gradient methods, such as PPO [26], has also been investigated in multi-agent RL. For example, [3] propose independent PPO (IPPO), a decentralized MARL, that can achieve high success rates in several hard SMAC maps. IPPO is, however, overall worse than QMIX [23], a method based on factorizing Q function to facilitate CTDE. Later methods based on factorized Q-learning include QTRAN [27] and QPLEX [30], where QPLEX has been shown to achive better performance than QMIX and QTRAN. Recently, [34] develop MAPPO, a PPO-based MARL algorithm that outperforms QMIX and QPLEX on some popular multi-agent environments such as SMAC [25, 4] and GRF [15]. To the best of our knowledge, MAPPO is currently a SOTA method for MARL. Our work integrates a new opponent imitation model into MAPPO, resulting in a new MARL algorithm that outperforms SOTA methods on various challenging game tasks.

**Imitation Learning (IL).**  In this study, we employ IL to anticipate the opponents' moves. IL is known as a compelling approach for sequential decision-making [20, 1]. In IL, a collection of expert trajectories is provided, with the objective of learning a policy that emulates behavior similar to the expert's policy. One of the simplest IL methods is Behavioral Cloning (BC), which aims to maximize the likelihood of the expert's actions under the learned policy. BC disregards environmental

dynamics, rendering it suitable only for uncomplicated environments. Several advanced IL techniques, encompassing environmental dynamics, have been proposed [24, 8, 12]. While these methods operate in complex and continuous domains, they involve adversarial learning, making them prone to instability and sensitivity to hyperparameters. The IQ-learn [9] stands as a cutting-edge IL algorithm with distinct advantages, specifically its incorporation of dynamics awareness and non-adversarial training. It's important to note that all the aforementioned IL methods were designed for single-agent RL. In contrast, the literature on multi-agent RL is limited, with only a handful of studies addressing IL in multi-agent RL. For instance, [28] presents an adversarial training-based algorithm, named Multi-agent Generative Adversarial IL. It's worth noting that all the IL algorithms mentioned above are established on the premise that expert (aka. opponent in our case) actions are either observable or can be accessed via sampling, which implies that no existing algorithm can be directly applied to our multi-agent game settings with *local state-only* observations.

**Opponent Modeling.** Many existing works in MARL attempt to capture the learning process of opponents and incorporate it into the learning of the agent's policy [6, 14, 33]. For example, the LOLA algorithm [6] considers the impact of one agent's policy on the parameter update of other opponents while Meta-MAPG [14] combines LOLA with meta-learning, accounting for continuous adaptation. Another important line of research on opponent modelling follows hierarchical reasoning, considering each agent holds a belief about the other agents according to varying levels of reasoning ability [32, 31, 35, 19]. As an example, in [32], they introduce a probabilistic recursive reasoning framework in which variational Bayes methods are used to approximate the opponents' conditional policies. Depart from these two lines of research, there are many other works that attempt to learn a representation for the opponent's policy or to consider the prediction of opponents' actions as an an auxiliary task that can be trained simultaneously with the RL part [11, 13, 22, 21, 10]. For example, [21] use variational encoder to model the opponents' fixed policies while [10] apply behavioral cloning together with agent identification to learn a hybrid generative-discriminative representation for the opponents' policy. All the aforementioned related works require having access to opponent's observations and actions during training and/or execution. In our work, on the other hand, the opponent modeling can be only trained based on local observations of each agent in the allied team. These local observations contain limited information about current states of nearby opponents while opponents' actions are unobservable. As a result, existing methods on opponent modeling are not applicable in our multi-agent setting.

## 3 Multi-Agent POMDP Setting

We consider a multi-player Markov game in which there are multiple agents forming an alliance to play against some opponent agents. We present the Markov game as a tuple $\{\mathcal{S}, \mathcal{N}_\alpha, \mathcal{N}_e, \mathcal{A}^\alpha, \mathcal{A}^e, P, R\}$, where $\mathcal{S}$ is the set of global states shared by all the agents, $\mathcal{N}_\alpha$ and $\mathcal{N}_e$ are the set of ally and enemy agents, $\mathcal{A}^\alpha = \prod_{i \in \mathcal{N}_\mathcal{A}} \mathcal{A}_i^\alpha$ is the set of joint actions and $\mathcal{A}^e = \prod_{j \in \mathcal{N}_e} \mathcal{A}_j^e$ is the set of joint actions of all the ally agents, $P$ is the transition dynamics of the game environment, and $R$ is a reward function that takes inputs as states and actions of all agents and returns the corresponding rewards. At each time step where the global state is $S$, each ally agent $i \in \mathcal{N}_\alpha$ makes an action $a_i^\alpha$ according to a policy $\pi_i^\alpha(a_i^\alpha | o_i^\alpha)$, where $o_i^\alpha$ is the observation of ally agent $i$ given state $S$. The joint action of allied agents can be now defined as $A^\alpha = \{a_i^\alpha \mid i \in \mathcal{N}_\alpha\}$, and the joint policy is defined accordingly:

$$\Pi^\alpha(A^\alpha | S) = \prod_{i \in \mathcal{N}^\alpha} \pi_i^\alpha(a_i^\alpha | o_i^\alpha).$$

The enemy agents, at the same time, make a joint action $A^e = \{a_j^e \mid j \in \mathcal{N}_e\}$ with the probability:

$$\Pi^e(A^e | S) = \prod_{j \in \mathcal{N}^e} \pi_j^e(a_j^e | o_j^e).$$

After all agents make decisions, the global state transits to a new state $S'$ with the probability $P(S' | A^e, A^\alpha, S)$. In our setting, the enemies' policies $\Pi^e$ are fixed and thus can be treated as a part of the environment dynamics, as follows:

$$P(S' | A^\alpha, S) = \sum_{A^e} \Pi(A^e | S) P(S' | A^e, A^\alpha, S)$$

Our goal is to find a policy that optimizes the allies' expected joint reward, formulated as follows:[1]

$$\max_{\Pi^\alpha} \mathbb{E}_{(A^\alpha, S) \sim \Pi^\alpha} \left[ R^\alpha(S, A^\alpha) \right] \tag{1}$$

---

[1]Environment dynamics are implicitly involved in sampling.

The game dynamics involve both the environment dynamics and the joint policy of enemies, making the training costly to converge. We aim to migrate uncertainties associated with these game dynamics by first predicting the opponent policy based on the allies' past observations and leveraging this prediction into guiding the policy training for the allies.

# 4 Opponent Policy Imitation

The key challenge in our problem is that actions taken by opponents are hidden from allied agents. Moreover, each allied agent has limited observations of other agents; they can only obtain information about nearby opponents. For example, in the SMAC environment, for each allied agent, besides information about the agent itself, the allied agent is also aware of the relative position and health point, etc. of the neighboring opponents.[2] Therefore, instead of directly predicting opponents' next moves, we focus on anticipating next states of opponents — this next-state prediction can be used as an implication of what actions have been taken by the neighboring opponents. Our key contributions include: (i) a novel representation of the opponent next-state prediction in the form of multi-agent IL; (ii) a new adaptation of IQ-Learn to solve our new IL problem; (iii) a comprehensive theoretical analysis on the influence of policy learning of allied agents on the next-state prediction outcomes; and (iv) a practical multi-agent IL algorithm which is tailored to local observations of allied agents.

Here, it is important to note that prior works on IL with state-only observations all assume that actions are not available in the expert demonstrations but can be accessed via sampling, which is not the case in our context. Alternatively, one could apply standard supervised learning for this opponent-next-state prediction task. However, a well-known drawback of this approach is that it disregards environment dynamics and often struggles with distribution shifts [16]. As shown later in our experiments, our IL approach significantly outperforms this supervised-learning approach.

## 4.1 Multi-Agent IL with Unobservable Actions

We now present our IL formulation and our adaptation of IQ-Learn for solving our new IL problem. For the sake of theoretical analysis, this section focuses on IL with *global* state-only observations. We then introduce a new practical algorithm later which addresses local observations of allied agents.

**Opponent Next-State Prediction as an IL.** To formulate the problem as an IL that accounts for the action-unobservable issue, we introduce a new notion of the "expert" state in our IL problem as a pair $W = (S, A^\alpha_-)$ which comprises of the original state $S$ and the joint action of the allies $A^\alpha_-$ taken in the previous step that leads to state $S$. The action space of the "expert" is equivalent to the original state space $\mathcal{S}$. We then introduce a new notion of a reward function for the expert as $R^e(W, S')$. This action ($S'$) of the expert is basically a resulting state of joint actions of the allies $A^\alpha$ and *hidden* joint actions of the enemies $A^e$ taken at state $S$. Altogether with the reward function $R^e(W, S')$, we introduce a new notion of joint policy for the expert, $\Pi^e(S'|W)$ (or $\Pi^e(S'|S, A^\alpha_-)$), which is essentially the probability of ending up at a global state $S'$ from state $S$. The dynamics in this IL setting becomes $P(W'|W, S') = P((S', A)|(S, A^\alpha_-), S') = \Pi^\alpha(A|S)$ (which is the allies' policy) where $W' = (S', A)$ and $A$ is the action taken by the allies at state $S$.

Let $\mathbf{\Pi} = \{\Pi : \mathcal{S} \times \mathcal{S} \times \mathcal{A}^\alpha \to [0, 1], \sum_{S' \in \mathcal{S}} \Pi(S'|S, A^\alpha_-) = 1, \forall S, S' \in \mathcal{S}, A^\alpha \in \mathcal{A}^\alpha\}$ be the support set of the imitating policy. We now introduce the maximum-entropy inverse RL framework [12] w.r.t the new notions of the expert's reward and policy $(R^e(S'|W), \Pi^e(S'|W))$:

$$\max_{R^e} \min_{\Pi \in \mathbf{\Pi}} \left\{ L(\Pi, R^e) = \mathbb{E}_{\rho^{e,\alpha}}[R^e(W, S')] - \mathbb{E}_{\rho^{\Pi,\alpha}}[R^e(W, S')] + \mathbb{E}_{\rho^{\alpha,\Pi}}[\ln \Pi(S'|W)] \right\} \quad (2)$$

where $\rho^{e,\alpha}$ is the occupancy measure of $(W, S')$ given by the expert policy $\Pi^e$ and the ally joint policy $\Pi^\alpha$, and $\rho^{\Pi,\alpha}$ the occupancy measure of $(W, S')$ given by the imitation and ally policies. In particular, $\rho^{e,\alpha}$ can be computed as follows:

$$\rho^{e,\alpha}(S', W) = (1 - \gamma)\Pi^e(S'|W)\prod_{t=0}^{\infty} \gamma^t P(W_t = W | \Pi^e, \Pi^\alpha)$$

**An Adaptation of IQ-Learn.** Drawing inspiration from the SOTA IL algorithm, IQ-learn, we construct our IL algorithm which is an adaptation of IQ-Learn tailored to our multi-agent environment.

---

[2]These neighbors change over time depending on actions of all agents and the environment dynamics.

The main idea of IQ-learn is to convert a reward learning problem into a Q-function learning one. To apply IQ-Learn to our setting, we present the following new *soft and inverse soft* Bellman operators, which is as adaption from the original ones introduced in [9]:

$$\mathcal{B}_{Q^e}^{\Pi, R^e}(W, S') = R^e(W, S') + \gamma \mathbb{E}_{W'}[V_{\Pi}^e(W')] \tag{3}$$

$$\mathcal{T}_{Q^e}^{\Pi}(W, S') = Q^e(W, S') - \gamma \mathbb{E}_{W'}[V_{\Pi}^e(W')] \tag{4}$$

where the state value function is computed as follows:

$$V_{\Pi}^e(W) = \mathbb{E}_{S' \sim \Pi}\big[Q^e(W, S') - \ln(\Pi(S'|W))\big]$$

First, it is clear that $\mathcal{B}_{Q^e}^{\Pi, R^e}$ is contractive, thus defining a unique fixed point solution $Q^*$ such that $\mathcal{B}_{Q^*}^{\Pi, R^e} = Q^*$. Let us further define the following function of $\Pi$ and $Q^e$ for the Q-function learning:

$$J(\Pi, Q^e) = \mathbb{E}_{\rho^e, \alpha}[\mathcal{T}_{Q^e}^{\Pi}(W, S')] - \mathbb{E}_{\rho^{\Pi, \alpha}}[\mathcal{T}_{Q^e}^{\Pi}(W, S')] + \mathbb{E}_{\rho^{\Pi, \alpha}}[\ln \Pi(S'|W)]$$

We obtain a theoretical result on a connection between the learning reward and learning Q-functions:[3]

**Proposition 1.** *For any reward function $R^e$, let $Q^*$ be the unique fixed point solution to the soft Bellman equation $\mathcal{B}_{Q^*}^{\Pi, R^e} = Q^*$, then: $L(\Pi, R^e) = J(\Pi, Q^*)$, and for any $Q^e$, $J(\Pi, Q^e) = L(\Pi, \mathcal{T}_{Q^e}^{\Pi})$.*

Proposition (1) indicates that the IL problem in (2) is equivalently to the Q-value-based IL problem:

$$\max_{Q^e} \min_{\Pi} J(\Pi, Q^e) \tag{5}$$

Suppose $Q^*$ is a solution to (5), then rewards can be recovered by taking $R^e(W, S') = Q^*(W, S') - \gamma \mathbb{E}_{W'}[V_{\Pi}^e(W')]$. Under this viewpoint, Prop. 2 shows that key properties of original IQ-learn still hold in our multi-agent setting with missing observations, making our IL algorithm convenient to use.

**Proposition 2.** *The IL problem (5) is equivalent to the maximization $\max_{Q^e} J(\Pi^Q, Q^e)$ where the imitation policy can be computed based on $Q^e$ as follows:*

$$\Pi^Q(S'|W) = \frac{\exp(Q^e(W, S'))}{\sum_{S''} \exp(Q^e(W, S''))}$$

*Moreover, the function $J(\Pi^Q, Q^e)$ is concave in $Q^e$.*

## 4.2 Affect of Allies' Policies on Imitation Learning

The above IL algorithm is trained during the training of the allies' policies, which means the dynamics $P(W'|W, S') = \Pi^{\alpha}(A^{\alpha}|S)$ change during the IL process. Therefore, we aim to analyze the impact of these changes on the imitation policy. According to Proposition 2, given $Q^e$, we can compute corresponding optimal imitation policy as $\Pi^Q$. Therefore, the value function $V_{\Pi}^e(W)$ can be alternatively write as $V_Q^e(W)$. We now can denote the loss function of the imitation model (Eq. 8) as a function of the allies' joint policy explicitly: $\Phi(\Pi^{\alpha}|Q^e) \equiv J(\Pi, Q^e)$.

We first present our results about bounds on the impact of the allies' changing policies on the IL loss function (Prop. 3). Based on these results, we then provide a bound on the IL learning outcome accordingly (Corollary 4). Let us denote by $\overline{Q} = \max_{(W, S')} Q^e(W, S')$ an upper bound of $Q^e$.

**Proposition 3.** *Given two allies' joint policies $\Pi^{\alpha}$ and $\widetilde{\Pi}^{\alpha}$ such that $\mathrm{KL}(\Pi^{\alpha}(\cdot|S)||\widetilde{\Pi}^{\alpha}(\cdot|S)) \leq \epsilon$ for any state $S \in \mathcal{S}$, the following inequality holds:*

$$\left|\Phi(\Pi^{\alpha}|Q^e) - \Phi(\widetilde{\Pi}^{\alpha}|Q^e)\right| \leq \left(\alpha \overline{Q} + \beta \ln |\mathcal{S}|\right) \sqrt{2 \ln 2\epsilon}$$

*where $\alpha = \frac{\gamma}{(1-\gamma)^2} + \frac{\gamma^2}{1-\gamma} + (3-\gamma)$, and $\beta = \frac{\gamma^2}{(1-\gamma)} + 3 - \gamma$.*

Let $\Phi(\Pi^{\alpha}|Q^e, \Pi)$ be the objective of IL $J(\Pi, Q^e)$, written as a function of the allies' joint policy $\Pi^{\alpha}$. The following proposition establishes a bound for the variation of $|\Phi(\Pi^{\alpha}|Q^e, \Pi) - \Phi(\widetilde{\Pi}^{\alpha}|Q^e, \Pi)|$ as function of $\mathrm{KL}(\Pi^{\alpha}||\widetilde{\Pi}^{\alpha})$, for any pair of allies' joint policies $(\Pi^{\alpha}, \widetilde{\Pi}^{\alpha})$.

Prop. 3 allows us to establish an upper bound for the IL when the allies' joint policy changes.

---

[3]All proofs are in the appendix.

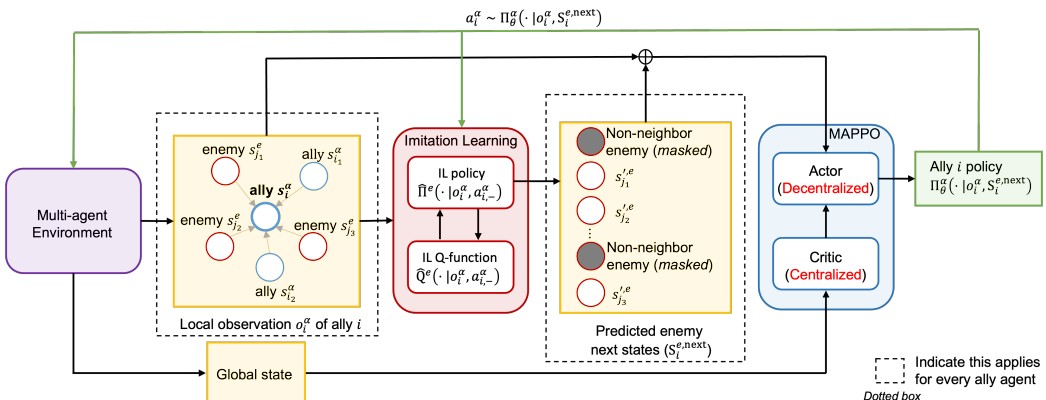

Figure 1: An overview of our IMAX-PPO algorithm. Each local observation $o_i^\alpha$ of an ally agent $i$ includes information about itself, as well as enemy and ally agents in its neighborhood (which changes over time). The output of the IL component is the predicted next states of neighboring enemy agents (predictions for the non-neighbor enemies will be masked out).

**Corollary 4.** *Given two allies' policies $\Pi^\alpha$ and $\widetilde{\Pi}^\alpha$ with $\mathrm{KL}(\Pi^\alpha(\cdot|S)||\widetilde{\Pi}^\alpha(\cdot|S)) \leq \epsilon$, $\forall S \in \mathcal{S}$, then:*

$$\left| \max_{Q^e} \left\{ \Phi(\Pi^\alpha|Q^e) \right\} - \max_{Q^e} \left\{ \Phi(\widetilde{\Pi}^\alpha|Q^e) \right\} \right| \leq \mathcal{O}(\sqrt{\epsilon})$$

Since the allies' joint policy $\Pi^\alpha$ will be changing during our policy learning process, the above result implies that the imitating policy will be stable if $\Pi^\alpha$ becomes stable, and if $\Pi^\alpha$ is converging to a target policy $\Pi^{\alpha*}$, then the imitator's policy also converges to the one that is trained with the target ally policy with a rate of $\sqrt{\mathrm{KL}(\Pi^\alpha||\widetilde{\Pi}^\alpha)}$. That is, if the actual policy is within a $\mathcal{O}(\epsilon)$ neighborhood of the target policy (i.e., $\mathrm{KL}(\Pi^\alpha||\widetilde{\Pi}^\alpha) \leq \epsilon$) then the expected return of the imitating policy is within a $\mathcal{O}(\sqrt{\epsilon})$ neighborhood of the desired "*expected return*" given by the target policy.

# 5 IMAX-PPO: Imitation-enhanced Multi-Agent EXtended PPO Algorithm

We present our MARL algorithm for the competive game setting. We first focus on a practical implementation of an IL algorithm taking into account local observations. We then show how to integrate this into our MARL algorithm. We call our algorithm as IMAX-PPO, standing for *Imitation-enhanced Multi-Agent EXtended PPO* algorithm.

## 5.1 Imitation Learning with Local Observations

In previous section, we present our new IL algorithm (which is an adaptation of IQ-Learn) to learn an expert policy $\widehat{\Pi}^e(S'|W) = \widehat{\Pi}^e(S'|S, A^\alpha)$ that behaves similarly to the probabilities of ending up at state $S'$ when the current global state is $S$ and the allies' joint action is $A^\alpha$. From the allies' perspective, to run this IL algorithm, it requires the allies to have access to the global state $S$. However, each ally agent $i$ can only observe local states of its neighboring enemies (such as their locations, speeds, etc.). Therefore, we adapt our IL algorithm in accordance with such local information. The goal is to predict next states of enemies in the neighborhood of each allied agent $i$, denoted by $\widehat{\Pi}^e(S_i^{e,\text{next}}|w_i^\alpha)$, where local information $w_i^\alpha = (o_i^\alpha, a_{i,-}^\alpha)$ with $o_i^\alpha$ is an observation vector of agent $i$, containing the local states of the agent $i$ itself and of all the agents in the neighborhood that are observable by agent $i$. In particular, given that $s_{i_k}^\alpha$ is the local state of an ally agent $i_k$ and $s_{j_k}^e$ is of an enemy agent $j_k$ and $N(i)$ is the neighborhood of $i$, we have:

$$o_i^\alpha = \{s_i^\alpha\} \cup \{s_{i_k}^\alpha : i_k \in N(i) \cap \mathcal{N}^\alpha\} \cup \{s_{j_k}^e : j_k \in N(i) \cap \mathcal{N}^e\}$$

To apply our IL algorithm to this local observation setting, we build a common policy network $\widehat{\Pi}_{\psi_\pi}^e$ and Q network $\widehat{Q}_{\psi_Q}^e$ for all the agents where $\psi_\pi$ and $\psi_Q$ are the network parameters. The IL objective

function can be reformulated according to local observations of the allies as follows:

$$J(\widehat{\Pi}^e_{\psi_\pi}, \widehat{Q}^e_{\psi_Q}) = \sum_{i \in \mathcal{N}^\alpha} \mathbb{E}_{(S_i^{e,\text{next}}, w_i^\alpha) \sim \rho^{e,\alpha}} \Big[ \widehat{Q}^e(S_i^{e,\text{next}}, w_i^\alpha) \tag{6}$$

$$- \gamma \mathbb{E}_{w_i^{\alpha,\text{next}}}[V_\Pi^e(w_i^{\alpha,\text{next}})]\Big] - (1-\gamma)\mathbb{E}_{w_{i0}^\alpha \sim P^0, \Pi^\alpha} V_\Pi^e(w_{i0}^\alpha)$$

where $w_i^\alpha = (o_i^\alpha, a_{i,-}^\alpha)$, $w_i^{\alpha,\text{next}} = (o_i^{\alpha,\text{next}}, a_i^\alpha)$ ($a_i^\alpha$ is the action taken by agent $i$ at observation $o_i^\alpha$, resulting in next observation $o_i^{\alpha,\text{next}}$), $S_i^{e,\text{next}} = \{s_{j_k}'^e : j_k \in N(i) \cap \mathcal{N}^e\}$ is the next states of enemies in the current neighborhood of the agent $i$. In addition, the value functions are re-formulated as follows:

$$V_\Pi^e(w_i^\alpha) = \mathbb{E}_{(S_i^{e,\text{next}}) \sim \widehat{\Pi}^e_{\psi_\pi}} \big[ \widehat{Q}^e_{\psi_Q}(S_i^{e,\text{next}}, w_i^\alpha) - \ln(\widehat{\Pi}^e_{\psi_\pi}(S_i^{e,\text{next}}|w_i^\alpha)) \big]$$

In the end, we can update $\widehat{\Pi}^e_{\psi_\pi}$ and $\widehat{Q}^e_{\psi_Q}$ by the following actor-critic rule: for a fixed $\widehat{Q}^e_{\psi_Q}$, we update $\psi_Q$ to maximize $J(\widehat{\Pi}^e_{\psi_\pi}, \widehat{Q}^e_{\psi_Q})$, and for a fixed $\widehat{\Pi}^e_{\psi_\pi}$, we apply soft actor-critic (SAC) to update $\psi_\pi$.

### 5.2 IMAX-PPO Algorithm

We now combine the local observation $o_i^\alpha$ of each allied agent $i$ with the next-state prediction $S_i^{e,\text{next}}$ of its neighboring enemies (obtained by our IL algorithm) to create an augmented input. This augmented input is used to improve the policy learning of the allied agent $i$. That is, we aim to optimize the allies' policy $\Pi_\theta^\alpha(a_i^\alpha|o_i^\alpha, S_i^{e,\text{next}}), i \in \mathcal{N}^\alpha\}$ that optimizes the long-term expected joint reward:

$$\max_{\Pi_\theta^\alpha} \mathbb{E}_{(a_i^\alpha, o_i^\alpha, S_i^{e,\text{next}}) \sim \Pi_\theta^\alpha} \Big[ \sum_{i \in \mathcal{N}^\alpha} R_i^\alpha(o_i^\alpha, a_i^\alpha) \Big]$$

where $o_i^\alpha$ is an observation vector of agent $i$, $S_i^{e,\text{next}}$ is the information derived from the imitator for agent $i$, $a_i^\alpha \in \mathcal{A}_i^\alpha$ is a local action of agent $i$. To facilitate the training and integration of the imitation learning policy into the MARL algorithm, for every ally agent $i$, we gather game trajectories following the structure $(o_i, a_i^\alpha, S_i^{e,\text{next}})$. These gathered observations are then stored in a replay buffer to train the imitation policy $\widehat{\Pi}^e_{\psi_\pi}(S_i^{e,\text{next}}|o_i^\alpha, a_i^\alpha)$.

In the IMAX-PPO framework, at each game state $S$, considering the current actor policy $\Pi^\alpha$ and the imitating policy $\widehat{\Pi}^e$, for each agent $i \in \mathcal{N}^\alpha$, we draw a sample for the allied agents' joint action $\widetilde{A}^\alpha \sim \Pi^\alpha$. Corresponding local observation $o_i^\alpha$ and action $\widetilde{a}_i^\alpha$ of each agent $i$ are then fed as inputs into the imitation policy to predict the subsequent state $S_i^{e,\text{next}} \sim \widehat{\Pi}^e(\cdot|o_i^\alpha, \widetilde{a}_i^\alpha)$. Once the predicted local states $\{S_i^{e,\text{next}}, i \in \mathcal{N}^\alpha\}$ are available, it is used as input to the actor policy $\Pi_\theta^\alpha$ in order to generate new actions for the allied agents. In simpler terms, we select a next local action $a_i'^\alpha \sim \Pi_\theta^\alpha(\cdot|o_i^\alpha, S_i^{e,\text{next}})$. Beside the allies' policy network, we also use a centralized value network $V_{\theta_v}^\alpha(S)$ and update it together with the policy network in an actor-critic manner, similarly to MAPPO. The actor-network is trained by optimizing the following objective:

$$L^\alpha(\theta) = \sum_{i \in \mathcal{N}^\alpha} \mathbb{E}_{o_i^\alpha, a_i^\alpha, S_i^{e,\text{next}}} \big[ \min\{r_i(\theta)\widehat{A}, \text{clip}(r_i(\theta), 1-\epsilon, 1+\epsilon)\widehat{A}\} \big] \tag{7}$$

where $r_i(\theta) = \frac{\Pi_\theta^\alpha(a_i^\alpha|o_i^\alpha, S_i^{e,\text{next}})}{\Pi_{\theta_{old}}^\alpha(a_i^\alpha|o_i^\alpha, S_i^{e,\text{next}})}$ and $\widehat{A}$ is the advantage function, calculated by Generalized Advantage Estimation (GAE). The Critic network is trained by optimizing

$$\Phi^\alpha(\theta_v) = \mathbb{E}_S \Big[ \max \Big\{ [V_{\theta_v}^\alpha(S) - \widehat{R}(S)]^2, [V_{\theta_v, \theta_{v,old}}^{\text{clip}}(S) - \widehat{R}(S)]^2 \Big\} \Big]$$

where $\widehat{R}(S) = \widehat{A} + V_{\theta_{v,old}}^\alpha(S)$ and $V_{\theta_v, \theta_{v,old}}^{\text{clip}}(S) = \text{clip}(V_{\theta_v}^\alpha(S), V_{\theta_{v,old}}^\alpha(S) - \epsilon, V_{\theta_{v,old}}^\alpha(S) + \epsilon)$. We provide the key stages in Algorithm 1. Additionally, Fig. 1 serves as an illustration of our IMAX-PPO.

## 6 Experiments

We evaluate the performance of our **IMAX-PPO** algorithm (Algo. 1) in comparison with some standard and SOTA multi-agent RL algorithms: **IPPO**, **MAPPO**, **QMIX** and **QPLEX**. In addition, to examine the impact of our multi-agent IL model on the performance of **IMAX-PPO**, we include two versions of **IMAX-PPO** where opponent's next states are predicted by (i) an adaption of the GAIL

**Algorithm 1** IMAX-PPO Algorithm

**Input**: Initial allies' policy network $\Pi_\theta^\alpha$, initial allies' value network $V_{\theta_v}^\alpha$, initial imitator's policy network $\widehat{\Pi}_{\psi_\pi}^e$, initial imitator's Q network $Q_{\psi_Q}^e$, learning rates $\kappa_\pi^e, \kappa_Q^e, \kappa_\pi^\alpha, \kappa_V^\alpha$.

**Output**: Trained allies' policy $\Pi_\theta^\alpha$

1: **for** $t = 0, 1, \dots$ **do**
2:     **# Updating imitator:**
3:     $\psi_{Q,t+1} = \psi_{Q,t} + \kappa_Q^e \nabla_{\psi_Q}[J(\psi_Q)]$ # Train Q function using the objective in (6)
4:     $\psi_{\pi,t+1} = \psi_{\pi,t} - \kappa_\pi^e \nabla_{\psi_\pi} \mathbb{E}_{S_i^{e,\text{next}}}[V_\Pi^e(S_i^{e,\text{next}})]$ # Update policy $\widehat{\Pi}_{\psi_\pi}^e$ (for continuous domains)
5:     **# Updating allies' policy:**
6:     $\theta_{t+1} = \theta_t + \kappa_\pi^\alpha \nabla_\theta L^\alpha(\theta)$     # Update allies' actor by maximizing $L^\alpha(\theta)$
7:     $\theta_{v,t+1} = \theta_{v,t} - \kappa_V^\alpha \nabla_{\theta_v} \Phi^\alpha(\theta_v)$ # Update allies' critic by minimizing $\Phi^\alpha(\theta_v)$
8: **end for**
9: **return** policy solution for allied agents

Table 1: Win-rates (percentage).

| Tasks | Scenarios | MAPPO | IPPO | QMIX | QPLEX | Sup MAPPO | IMAX-PPO GAIL | InQ |
|---|---|---|---|---|---|---|---|---|
| SMAC Protoss | 5_vs_5 | 58.0 | 54.6 | 70.2 | 53.3 | 71.8 | 68.1 | **78.7** |
| | 10_vs_10 | 58.3 | 58.0 | 69.0 | 53.7 | 67.3 | 59.6 | **79.8** |
| | 10_vs_11 | 18.2 | 20.3 | 42.5 | 22.8 | 36.7 | 21.3 | **48.7** |
| | 20_vs_20 | 38.1 | 44.5 | 69.7 | 27.2 | 71.1 | 76.3 | **80.6** |
| | 20_vs_23 | 5.1 | 4.1 | 16.5 | 4.8 | 21.9 | 11.8 | **24.2** |
| SMAC Terran | 5_vs_5 | 52.0 | 56.2 | 58.4 | **70.0** | 55.8 | 53.3 | 69.9 |
| | 10_vs_10 | 58.1 | 57.3 | 65.8 | 66.1 | 54.1 | 58.4 | **72.2** |
| | 10_vs_11 | 28.6 | 31.0 | 39.4 | 41.4 | 26.9 | 28.4 | **53.9** |
| | 20_vs_20 | 52.8 | 49.6 | 57.6 | 23.9 | 38.6 | 35.9 | **65.4** |
| | 20_vs_23 | 11.2 | 10.0 | 10.0 | 7.0 | 11.2 | 4.7 | **17.7** |
| SMAC Zerg | 5_vs_5 | 41.0 | 37.2 | 37.2 | 47.8 | 52.5 | 48.6 | **55.0** |
| | 10_vs_10 | 39.1 | 49.4 | 40.8 | 41.6 | 57.4 | 50.6 | **57.6** |
| | 10_vs_11 | 31.2 | 26.0 | 28.0 | 31.1 | 38.1 | 34.8 | **41.5** |
| | 20_vs_20 | 31.9 | 31.2 | 30.4 | 15.8 | **44.3** | 26.7 | 43.3 |
| | 20_vs_23 | 15.8 | 8.3 | 10.1 | 6.7 | 13.6 | 8.2 | **21.3** |
| Gold Miner | easy | 48.9 | 49.3 | 57.2 | 59.8 | 47.1 | 54.5 | **61.8** |
| | medium | 40.6 | 39.5 | 47.3 | 50.4 | 39.4 | 39.3 | **55.0** |
| | hard | 31.2 | 31.2 | 41.7 | 43.5 | 31.3 | 29.7 | **49.8** |
| GRF | 3_vs_1 | 88.0 | 82.7 | 8.1 | 90.2 | 96.1 | 96.4 | **98.1** |
| | easy | 87.8 | 84.1 | 16.0 | 94.9 | 89.7 | 64.1 | **95.0** |
| | hard | 77.4 | 70.9 | 3.2 | 95.1 | 10.7 | 15.2 | **97.3** |

algorithm [12], denoted as **IMAX-PPO (GAIL)** and (ii) the IQ-learn adaption (i.e. Algorithm 1, denoted as **IMAX-PPO (InQ)**. Moreover, to compare out IL-based approach with supervise learning, we include an approach based on MAPPO where the opponent's next states are learned and predicted by *standard supervising learning*. We denote this approach as **Sup-MAPPO.**

The details of **Sup-MAPPO** and **IMAX-PPO (GAIL)** are provided in the appendix. We run extensive experiments in three multi-agent competitive environments: SMACv2, Google Research Football (GRF), and Miner. Each reported value is computed based on 32 different rounds of game playing (each corresponds to a different random seed).

**SMACv2.** SMACv2 [4] is an advanced variant of SMAC, driven by the aim to present a more challenging setting for the assessment of cooperative MARL algorithms. In SMACv2, scenarios are procedurally generated, which require agents to generalize to previously unseen settings (from the same distribution) during evaluation. This benchmark consists of 15 sub-tasks where the number of agents varies from 5 to 20. The agents can play with opponents of different difficulty levels. In

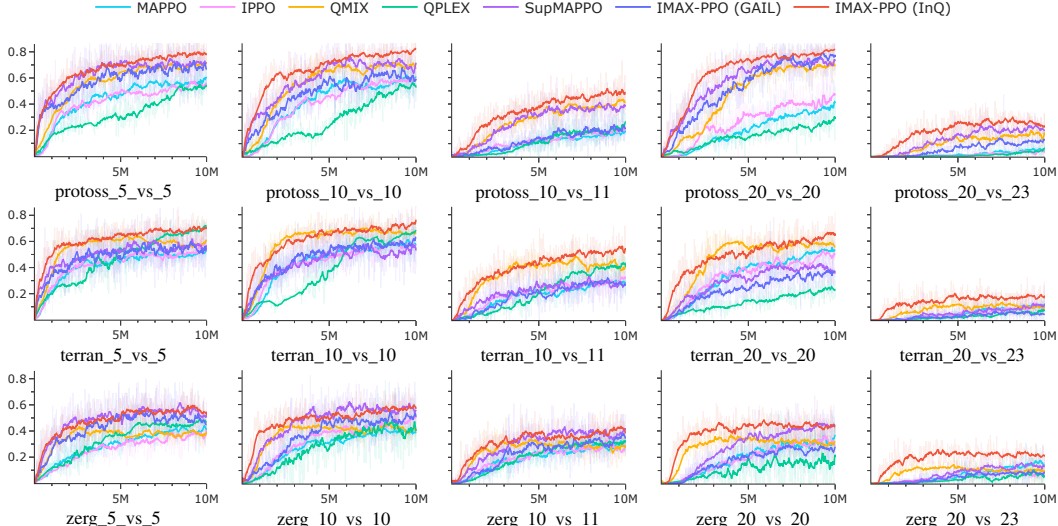

Figure 2: Win-rate curves on SMACv2 environment.

comparison to SMACv1 [25], SMACv2 stands apart by permitting randomized team compositions, varied starting positions, and an emphasis on augmenting diversity.

Figure 2 shows the performance of the five algorithms during the training process across 15 sub-tasks. The x-axis is the number of training steps and the y-axis is the winning rates averaged over 32 rounds of evaluations. In Figure 2, **IMAX-PPO (InQ)** consistently and significantly outperforms other baselines. Our algorithm frequently attains quicker convergence; it achieves high win rates at earlier training stages. This could be attributed to the incorporation of our IL component, which facilitates faster comprehension of opponents throughout the game. In particular, the **IMAX-PPO (InQ)** outperforms the two other variants **IMAX-PPO (GAIL)** and **Sup-MAPPO**, indicating the advantage of our inverse-Q approach over other IL (i.e. GAIL) and traditional supervising learning methods. Details of win rates at the end of training are shown in Table 1.

**Google Research Football (GRF).** This is a challenge on Kaggle competitions made by Google Research team [15]. We focus on three main sub-tasks, sorted based on increasing difficulty levels: (i) *academy-3-vs-1-with-keeper*: three allies try to score against a goal-keeping opponent; (ii) *academy-counterattack-easy*: four allies versus a counter-attack opponent and a goal-keeping opponent; and (iii) *academy-counterattack-hard*: four allies versus two counter-attack opponents and a goalkeeper.

By default, the representations of all agents' observations are RGB pixels in GRF, so we pre-process this information by distilling some important features such as object positions, object directions, distances between objects, etc. The final win rates are in Table 1, which shows that **IMAX-PPO (InQ)** achieves nearly 100% win-rates, and significantly outperforms other baselines.[4]

**Gold Miner [7].** This is another competitive multi-agent game for evaluating our methods, originating from a MARL competition. Multiple miners navigate in a 2D terrain containing obstacles and repositories of gold. Players get points according to the volume of gold they successfully extract. This game is challenging to win as the agents have to learn playing against extremely well-designed heuristic-based enemies. In this game, the ally agents win if the allied team's average mined gold is higher than that of the enemy team.

We customized the original environment into three sub-tasks (between two allies against two enemies) of three difficulty levels: (i) *Easy (easy_2_vs_2)*: The enemies' greedy strategy is to find the shortest way to the golds; (ii) *Medium (medium_2_vs_2)*: One enemy is greedy, and the other follows the algorithm of the second-ranking team in the competition; and (iii) *Hard (hard_2_vs_2)*: The enemies are the first- and second-ranking teams in the competition.

---

[4]The win rate curves during the training for GRF and Gold Miner environments are in the appendix.

For this environment, the win rates are in Table 1. Again, **IMAX-PPO (InQ)** obtained superior win rates across all three tasks. Especially, in the hard-level task, our algorithm manages to win more than $50\%$ of the time against the first and second-ranking teams in the competition.

## 7  Conclusion

We introduced a novel principled framework for enhancing agent training in multi-agent environments through IL. Our new IL model, adapted from IQ-learn, can predict opponents' policy using only local state observations. By integrating this model into a multi-agent PPO algorithm, our IMAX-PPO algorithm consistently outperforms previous SOTA algorithms such as QMIX and MAPPO. This improvement is observed across various challenging multi-agent tasks, including SMACv2 and GRF. A possible **limitation** of our work is that it relies on the assumption that the enemies do not update their policies during training (even though this is a standard setting in cooperative multi-agent reinforcement learning [4, 25]). A future direction would be to delve into these aspects to develop efficient MARL algorithms for cooperative-competitive multi-agent RL.

## Acknowledgments

This research/project is supported by the National Research Foundation Singapore and DSO National Laboratories under the AI Singapore Programme (AISG Award No: AISG2-RP-2020-017).

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

# Appendices

## A    Missing Proofs

### A.1    Proof of Proposition 1

**Proposition 1:**  *For any reward function $R^e$, let $Q^*$ be the unique fixed point solution to the soft Bellman equation $\mathcal{B}_{Q^*}^{\Pi, R^e} = Q^*$, then we have $L(\Pi, R^e) = J(\Pi, Q^*)$, and for any $Q^e$, $J(\Pi, Q^e) = L(\Pi, \mathcal{T}_{Q^e}^{\Pi})$.*

*Proof.*  The proof is similar to that given in [9], as we can see that if $Q^*$ is a solution to the soft Bellman equation, then

$$Q^*(W, S') = R^e(W, S') + \gamma \mathbb{E}_{W'}[V_{\Pi}^*(W')]$$

where

$$V_{\Pi}^*(W) = \mathbb{E}_{S' \sim \Pi}[Q^*(W, S') - \ln(\Pi(S'|W))]$$

which implies

$$R^e(W, S') = Q^*(W, S') - \gamma \mathbb{E}_{W'}[V_{\Pi}^*(W')] = \mathcal{T}_{Q^*}^{\Pi}(W, S')$$

which validates $L(\Pi, R^e) = J(\Pi, Q^*)$. The second inequality is just a trivial result from the definition of $J$ and $L$.  $\square$

### A.2    Proposition 5

Our following Proposition 5 shows that the function $J(\Pi, Q^e)$ can be reformulated in a more compact form that is convenient for training.

**Proposition 5.**  *The function $J(\cdot)$ can be written as follows:*

$$J(\Pi, Q^e) = \mathbb{E}_{(W, S') \sim \rho^{e, \alpha}}\left[Q^e(W, S') - \gamma \mathbb{E}_{W'}[V_{\Pi}^e(W')]\right]$$
$$- (1 - \gamma) \mathbb{E}_{W_0 \sim (P^0, \Pi^\alpha)} V_{\Pi}^e(W_0) \tag{8}$$

*where $S_0$ is an initial state and $P^0$ is the initial state distribution in the original MDP.*

*Proof.*  From the definition of $J(\Pi, Q^e)$ , we write function $J$ as

$$J(\Pi, Q^e) = \mathbb{E}_{\rho^{e, \alpha}}[\mathcal{T}_{Q^e}^{\Pi}(W, S')] - \mathbb{E}_{\rho^{\alpha, \Pi}}[\mathcal{T}_{Q^e}^{\Pi}(W, S')] + \mathbb{E}_{\rho^{\alpha, \Pi}}[\ln \Pi(S'|W)]$$
$$= \mathbb{E}_{(S, A_-^\alpha, S) \sim \rho^{e, \alpha}}[Q^e(W, S') - \gamma \mathbb{E}_{W'} V_{\Pi}^e(W')] - \mathbb{E}_{(S, A_-^\alpha, S') \sim \rho^{\alpha, \Pi}}[Q^e(W, S') - \gamma \mathbb{E}_{W'}[V_{\Pi}^e(W')]]$$
$$+ \mathbb{E}_{\rho^{\alpha, \Pi}}[\ln \Pi(S'|W)] \tag{9}$$

We consider the second and third terms of (9) and write

$$\mathbb{E}_{(S, A_-^\alpha, S') \sim \rho^{\alpha, \Pi}}[Q^e(W, S') - \gamma \mathbb{E}_{W'}[V_{\Pi}^e(W')]] - \mathbb{E}_{\rho^{\alpha, \Pi}}[\ln \Pi(S'|W)]$$

$$= (1 - \gamma)\left(\mathbb{E}_{S_t, A_{t-1}^\alpha, S_{t+1} \sim \Pi^\alpha, \Pi}\left[\sum_{t=0} \gamma^t Q^e(S_{t+1}, W_t)\right] - \gamma\left(\mathbb{E}_{W_t \sim \Pi^A, \Pi}\left[\sum_{t=0} \gamma^t V_{\Pi}^e(W_{t+1})\right]\right)\right.$$
$$\left. - \mathbb{E}_{S_t, A_{t-1}^\alpha, S_{t+1} \sim \Pi}\left[\sum_{t=0} \gamma^t \ln \Pi(S_{t+1}|W_t)\right]\right)$$

$$= (1 - \gamma)\left(\mathbb{E}_{S_t, A_{t-1}^\alpha, S_{t+1} \sim \Pi^\alpha, \Pi}\left[\sum_{t=0} \gamma^t Q^e(S_{t+1}, W_t)\right] - \gamma\left(\mathbb{E}_{S_{t+1}, S_t, A_{t-1}^\alpha \sim \Pi^\alpha, \Pi}\left[\sum_{t=1} \gamma^{t-1} Q^e(S_{t+1}, W_t)\right]\right)\right.$$
$$\left. - \gamma\left(\mathbb{E}_{S_{t+1}, S_t, A_{t-1}^\alpha \sim \Pi^\alpha, \Pi}\left[\sum_{t=1} \gamma^{t-1} \ln \Pi(S_{t+1}|W_t)\right]\right) + \mathbb{E}_{S_t, A_{t-1}^\alpha, S_{t+1} \sim \Pi^\alpha, \Pi}\left[\sum_{t=0} \gamma^t \ln \Pi(S_{t+1}|W_t)\right]\right)$$
$$= (1 - \gamma) \mathbb{E}_{S_1, W_0}[Q^e(S_1, W_0) - \ln \Pi(S_1|W_0)]$$
$$= (1 - \gamma) \mathbb{E}_{W_0} V_{\Pi}^e(W_0). \tag{10}$$

Thus,

$$J(\Pi, Q^e) = \mathbb{E}_{(S, A_-^\alpha, S') \sim \rho^{e, \alpha}}[Q^e(W, S') - \gamma \mathbb{E}_{W'} V_{\Pi}^e(W')] - (1 - \gamma) \mathbb{E}_{W_0 \sim P^0} V_{\Pi}^e(W_0) \tag{11}$$

as desired.  $\square$

## A.3 Proof of Proposition 2

**Proposition 2:** *The problem* $\max_{Q^e} \min_{\Pi} J(\Pi, Q^e)$ *is equivalent to the maximization* $\max_{Q^e} J(\Pi^Q, Q^e)$ *where*

$$\Pi^Q(S'|W) = \frac{\exp(Q^e(W, S'))}{\sum_{S''} \exp(Q^e(W, S''))}$$

*Moreover, the function* $J(\Pi^Q, Q^e)$ *is concave in* $Q^e$.

*Proof.* To simplify the proof, we first consider the following simpler optimization problem. Let $p_1, p_2, ..., p_N \in [0, 1]$ and $x_1, ..., x_N$ are N real numbers. Consider the maximization problem

$$\max_{p \in [0,1]^N} \sum_{i=1}^{N} x_i p_i - p_i \ln p_i \qquad \text{(P1)}$$

$$\text{subject to} \quad \sum_i p_i = 1$$

Using the KKT condition, if $p^*$ is an optimal solution to the above convex problem, $p^*$ needs to satisfy the following conditions: there exists $\lambda$ such that

$$\begin{cases} x_i - \ln p_i^* - 1 - \lambda = 0, \ \forall i \\ \sum_i p_i^* = 1 \end{cases}$$

which implies

$$p_i^* = \exp(x_i - 1 - \lambda)$$

Combine this with the condition $\sum_i p_i^* = 1$ we should have $\exp(-1 - \lambda) = \sum_i \exp(x_i)$ and $p_i^* = \frac{\exp(x_i)}{\sum_j \exp(x_j)}$ (*). In addition, when $p = p^*$, the objective function of (P1) can be written as

$$f(x) = \sum_i x_i p_i^* - p_i^* \ln(p_i^*)$$

$$= \sum_i p_i \left( x_i - \ln(p_i^*) \right)$$

$$= \sum_i p_i \left( x_i - x_i + \ln \left( \sum_i \exp(x_i) \right) \right)$$

$$= \ln \left( \sum_i \exp(x_i) \right)$$

We then see that $f(x)$ has a log-sum-exp form, thus it is convex in $x$ (**).

We now return to the minimax problem $\max_{Q^e} \min_{\Pi} J(\Pi, Q^e)$. For any fixed $Q^e$, it can be seen that the problem $\min_{\Pi} J(\Pi, Q^e)$ is equivalent to

$$\max_{\Pi} \left\{ \gamma \mathbb{E}_W[V_{\Pi}^e(W)] + (1 - \gamma) \mathbb{E}_{W_0 \sim P^0}[V_{\Pi}^e(W_0)] \right\}$$

We first consider the problem $\max_{\Pi} V_{\Pi}^e(W)$, recalling that $V_{\Pi}^e(W)$ can be written as

$$V_{\Pi}^e(W) = \sum_{S'} \Pi(S'|W) \times [Q^e(W, S') - \log(\Pi(S'|W))]$$

Then it can be seen that each term $\sum_{S'} \Pi(S'|W) \times [Q^e(W, S') - \log(\Pi(S'|W))]$ has the form of (P1), thus from (**) we see that $V_{\Pi}^e(S)$ is maximized at

$$\Pi^Q(S'|W) = \frac{\exp(Q^e(W, S'))}{\sum_{S''} \exp(Q^e(W, S''))}$$

Moreover, according to the result (**) proved above, when $\Pi = \Pi^Q$, $V_{\Pi}^e(W)$ is convex in $Q^e$. Consequently, the loss function of the IQ-learn

$$J(\Pi, Q^e) = \mathbb{E}_{(S, A_-^\alpha, S') \sim (\rho^e, \alpha)}[Q^e(W, S') - \gamma \mathbb{E}_{W'} V_{\Pi}^e(W')] + (1 - \gamma) \mathbb{E}_{W^0 \sim P^0} V_{\Pi}^e(W^0)$$

is concave in $Q^e$, which completes the proof. $\qquad \square$

## A.4 Corollary 6

**Corollary 6.** *If* $\Pi = \Pi^Q$ *(defined in Proposition 2), then we can write* $V^e_{\Pi}(S)$ *as follows:*

$$V^e_{\Pi}(W) = V^e_Q(W) = \ln\left(\sum_{S'} \exp\left(Q^e(W, S')\right)\right)$$

Note that Corollary 6 is just a direct result from the proof of Proposition 2.

## A.5 Proof of Proposition 3

**Proposition 3:** *Given two allies' joint policies* $\Pi^\alpha$ *and* $\widetilde{\Pi}^\alpha$ *such that* $\mathrm{KL}(\Pi^\alpha(\cdot|S)||\widetilde{\Pi}^\alpha(\cdot|S)) \leq \epsilon$ *for any state* $S \in \mathcal{S}$, *the following inequality holds:*

$$\left|\Phi(\Pi^\alpha|Q^e) - \Phi(\widetilde{\Pi}^\alpha|Q^e)\right| \leq \left(\alpha\overline{Q} + \beta\ln|\mathcal{S}|\right)\sqrt{2\ln 2\epsilon}$$

*where* $\overline{Q} = \max\limits_{(W,S')} Q^e(W, S')$ *is an upper bound of* $Q^e$, $\alpha = \frac{\gamma}{(1-\gamma)^2} + \frac{\gamma^2}{1-\gamma} + (3 - \gamma)$, *and* $\beta = \frac{\gamma^2}{(1-\gamma)} + 3 - \gamma..$

*Proof.* We first write the loss function as

$$\Phi(\Pi^\alpha|Q^e) = \mathbb{E}_{(S,A^\alpha_-,S')\sim(\rho^e,\alpha)}\left[Q^e(W,S') - \gamma\mathbb{E}_{W'\sim\Pi^\alpha}V^e_Q(W')\right] - (1-\gamma)\mathbb{E}_{W^0\sim P^0,\Pi^\alpha}V^e_Q(W_0)$$

We first write the difference between two loss values as

$$\Phi(\Pi^\alpha|Q^e) - \Phi(\widetilde{\Pi}^\alpha|Q^e) = \mathbb{E}_{(S,A^\alpha_-,S')\sim\rho^{e,\alpha}}[Q^e(W,S')] - \mathbb{E}_{(S,A^\alpha_-,S')\sim\rho^{e,\tilde{\alpha}}}[Q^e(W,S')]$$
$$-\gamma\left(\mathbb{E}_{W'\sim\rho^{e,\alpha}}[V^e_Q(W')] - \mathbb{E}_{W'\sim\rho^{e,\tilde{\alpha}}}[V^e_Q(W')]\right)$$
$$-(1-\gamma)\left(\mathbb{E}_{W_0\sim P^0,\Pi^\alpha}(V^e_Q(W_0) - V^e_Q(W_0))\right) \tag{12}$$

We first consider the first term of (12). Let us denote the following function

$$\Gamma(S'|W) = \mathbb{E}_{(S'_t,W_t)\sim\Pi^e,\widetilde{\Pi}^\alpha}\left[\sum_{t=0}\gamma^t\left(Q^e(S'_t,W_t)\right)\Big|(S'_0,W_0) = (W,S')\right]$$

and write the first term of (12) as

$$(1-\gamma)\left(\mathbb{E}_{(S'_t,W_t)\sim(\Pi^e,\Pi^\alpha)}\left[\sum_{t=0}\gamma^t\left(Q^e(S'_t,W_t)\right)\right] - \mathbb{E}_{(S'_t,W_t)\sim(\Pi^e,\widetilde{\Pi}^\alpha)}\left[\sum_{t=0}\gamma^t\left(Q^e(S'_t,W_t)\right)\right]\right)$$

$$= (1-\gamma)\left(\mathbb{E}_{(S'_t,W_t)\sim(\Pi^e,\Pi^\alpha)}\left[\sum_{t=0}\gamma^t\left(Q^e(S'_t,W_t)\right)\right] - \mathbb{E}_{(S'_0,W_0)\sim(P^0,\widetilde{\Pi}^\alpha)}\left[\Gamma(S'_0|W_0)\right]\right)$$

$$= (1-\gamma)\left(\mathbb{E}_{(S'_t,W_t)\sim(\Pi^e,\Pi^\alpha)}\left[\sum_{t=0}\gamma^t\left(Q^e(S'_t,W_t)\right)\right] - \mathbb{E}_{(S'_0,W_0)\sim(P^0,\Pi^\alpha)}\left[\Gamma(S'_0|W_0)\right]\right)$$

$$+ (1-\gamma)\left(\mathbb{E}_{(S'_0,W_0)\sim(P^0,\Pi^\alpha)}\left[\Gamma(S'_0|W_0) - \mathbb{E}_{(S'_0,W_0)\sim(P^0,\widetilde{\Pi}^\alpha)}\left[\Gamma(S'_0|W_0)\right]\right]\right)$$

$$= (1-\gamma)\left(\mathbb{E}_{(S'_t,W_t)\sim(\Pi^e,\Pi^\alpha)}\left[\sum_{t=0}\gamma^t\left(Q^e(S'_t,W_t)\right)\right] + \mathbb{E}_{(S'_t,W_t)\sim(\Pi^e,\Pi^\alpha)}\left[\sum_{t=0}\gamma^t\left(\gamma\Gamma(S'_{t+1}|W_{t+1}) - \Gamma(S'_t|W_t)\right)\right]\right)$$

$$+ (1-\gamma)\left(\mathbb{E}_{(S'_0,W_0)\sim(P^0,\Pi^\alpha)}\left[\Gamma(S'_0|W_0) - \mathbb{E}_{(S'_0,W_0)\sim(P^0,\widetilde{\Pi}^\alpha)}\left[\Gamma(S'_0|W_0)\right]\right]\right)$$

$$= (1-\gamma)\mathbb{E}_{(S'_t,W_t)\sim(\Pi^e,\Pi^\alpha)}\left[\sum_{t=0}\gamma^t\left(Q^e(S'_t,W_t) + \gamma\Gamma(S'_{t+1}|W_{t+1}) - \Gamma(S'_t|W_t)\right)\right]$$

$$+ (1-\gamma)\left(\mathbb{E}_{(S'_0,W_0)\sim(P^0,\Pi^\alpha)}\left[\Gamma(S'_0|W_0) - \mathbb{E}_{(S'_0,W_0)\sim(P^0,\widetilde{\Pi}^\alpha)}\left[\Gamma(S'_0|W_0)\right]\right]\right) \tag{13}$$

We first see that

$$\Gamma(S|W) = \mathbb{E}_{(S'_t,W_t)\sim\Pi^e,\widetilde{\Pi}^\alpha}\left[\sum_{t=0}\gamma^t\left(Q^e(S'_t,W_t)\right)\Big|(S'_0,W_0) = (W,S')\right] \leq \frac{\overline{Q}}{1-\gamma}$$

Thus, we can bound the second term of (13) as

$$(1-\gamma)\left(\mathbb{E}_{(S_0',W_0)\sim(P^0,\Pi^\alpha)}\left[\Gamma(S_0'|W_0)\right]-\mathbb{E}_{(S_0',W_0)\sim(P^0,\widetilde{\Pi}^\alpha)}\left[\Gamma(S_0'|W_0)\right]\right)$$

$$\leq (1-\gamma)(\max_{S,W}\Gamma(S|W))||\Pi^\alpha(\cdot|S_t)-\widetilde{\Pi}^\alpha(\cdot|S_t)||_\infty$$

$$\leq \overline{Q}\max_S\left\{\sqrt{2\ln 2\mathbf{KL}(\Pi^\alpha(\cdot|S)||\widetilde{\Pi}^\alpha(\cdot|S))}\right\} \tag{14}$$

We also see that

$$\Gamma(S_t'|W_t)=Q^e(S_t',W_t)+\gamma\mathbb{E}_{(S_{t+1}',W_{t+1})\sim\Pi^e,\widetilde{\Pi}^\alpha|S_t',W_t}\left[\Gamma(S_{t+1}'|W_{t+1})\right]$$

Thus, we further can bound the first term (13) as

$$\mathbb{E}_{(S_{t+1}',W_{t+1})\sim\Pi^e,\Pi^\alpha|S_t',W_t}\left[\left(Q^e(S_t',W_t)+\gamma\Gamma(S_{t+1}'|W_{t+1})-\Gamma(S_t'|W_t)\right)\right]$$

$$=\gamma\mathbb{E}_{(S_{t+1}',W_{t+1})\sim\Pi^e,\Pi^\alpha|S_t',W_t}\left[\left(\Gamma(S_{t+1}'|W_{t+1})\right)\right]-\gamma\mathbb{E}_{(S_{t+1}',W_{t+1})\sim\Pi^e,\widetilde{\Pi}^\alpha|S_t',W_t}\left[\left(\Gamma(S_{t+1}'|W_{t+1})\right)\right]$$

$$\leq \gamma\mathcal{H}||\Pi^\alpha(\cdot|S_t)-\widetilde{\Pi}^\alpha(\cdot|S_t)||_\infty$$

$$\leq \gamma\mathcal{H}\max_S\left\{\sqrt{2\ln 2\mathbf{KL}(\Pi^\alpha(\cdot|S)||\widetilde{\Pi}^\alpha(\cdot|S))}\right\}$$

where:

$$\mathcal{H}=\max_{(W,S')}\left\{\Gamma(S'|W)\right\}$$

$$=\max_{(W,S')}\left\{\mathbb{E}_{(S_t',W_t)\sim(\Pi^e,\widetilde{\Pi}^\alpha)}\left[\sum_{t=0}\gamma^t\left(Q^e(S_t',W_t)\right)\right]\right\}$$

$$\leq \frac{\overline{Q}}{1-\gamma}$$

where $\overline{Q}$ is an upper bound of $Q^e(\cdot|\cdot)$. Therefore, we can bound (13) as

$$\mathbb{E}_{(S_t'|S_t,A_t^\alpha)\sim(\Pi^e,\Pi^\alpha)}\left[\sum_{t=0}\gamma^t\left(Q^e(S_t',W_t)+\gamma\Gamma(S_{t+1}'|W_{t+1})-\Gamma(S_t'|W_t)\right)\right]$$

$$+(1-\gamma)\left(\mathbb{E}_{(S_0',W_0)\sim(P^0,\Pi^\alpha)}\left[\Gamma(S_0'|W_0)-\mathbb{E}_{(S_0',W_0)\sim(P^0,\widetilde{\Pi}^\alpha)}\left[\Gamma(S_0'|W_0)\right]\right]\right) \tag{15}$$

$$\leq \left(\frac{\gamma}{(1-\gamma)^2}+1\right)\overline{Q}\max_S\left\{\sqrt{2\ln 2\mathbf{KL}(\Pi^\alpha(\cdot|S)||\widetilde{\Pi}^\alpha(\cdot|S))}\right\} \tag{16}$$

For the second term of (12), we can bound it as

$$\gamma\left(\mathbb{E}_{(W,S',W')\sim\rho^{e,\alpha}}\left[\gamma V_Q^e(W')\right]-\gamma\mathbb{E}_{(W,S',W')\sim\rho^{e,\tilde\alpha}}\left[\gamma V_Q^e(W')\right]\right)$$

$$=(1-\gamma)\left(\mathbb{E}_{W_t\sim\Pi^e,\Pi^\alpha}\left[\sum_{t=1}\gamma^t V_Q^e(W_t)\right]-\mathbb{E}_{(W_t)\sim\Pi^e,\widetilde{\Pi}^\alpha}\left[\sum_{t=1}\gamma^t V_Q^e(W_t)\right]\right)$$

$$=(1-\gamma)\mathbb{E}_{(W_t\sim\Pi^e,\Pi^\alpha)}\left[\sum_{t=1}\gamma^t\left(V_Q^e(W_t)+\gamma U(W_{t+1})-U(W_t)\right)\right]$$

$$+(1-\gamma)\left(\mathbb{E}_{(W_1)\sim(P^0,\Pi^\alpha)}[U(W_1)]-\mathbb{E}_{W_1\sim(P^0,\widetilde{\Pi}^\alpha)}[U(W_1)]\right) \tag{17}$$

where $U(W)=\mathbb{E}_{(W_t\sim\Pi^e,\widetilde{\Pi}^\alpha)}\left[\sum_{t=0}\gamma^t\left(V_Q^e(W_t)\right)\,\Big|\,W_0=W\right]$. It then can be seen that

$$U(W_t)=V_Q^e(W_t)+\gamma\mathbb{E}_{(W_{t+1})\sim\Pi^e,\widetilde{\Pi}^\alpha|W_t}[U(W_{t+1})] \tag{18}$$

and

$$U(W) \leq \frac{1}{1-\gamma} \max_W V_Q^e(W)$$

$$\leq \frac{1}{1-\gamma} \max_W \left\{ \ln \left( \sum_{S'} \exp(Q^e(W, S')) \right) \right\}$$

$$\leq \frac{1}{1-\gamma} (\ln |\mathcal{S}| + \overline{Q})$$

Thus, given any $W_t$, we have

$$\mathbb{E}_{W_{t+1} \sim (\Pi^e, \Pi^\alpha)|S_t} \left[ V_Q^e(W_t) + \gamma U(W_{t+1}) - U(W_t) \right]$$

$$= \gamma \mathbb{E}_{W_{t+1} \sim (\Pi^e, \Pi^\alpha)|W_t} [U(W_{t+1})] - \gamma \mathbb{E}_{W_{t+1} \sim (\Pi^e, \widetilde{\Pi}^\alpha)|W_t} [U(W_{t+1})]$$

$$\leq \gamma \mathcal{K} ||\Pi^\alpha(\cdot|S_t) - \widetilde{\Pi}^\alpha(\cdot|S_t)||_\infty$$

$$\leq \gamma \mathcal{K} \max_S \left\{ \sqrt{2 \ln 2 \mathbf{KL}(\Pi^\alpha(\cdot|S)||\widetilde{\Pi}^\alpha(\cdot|S))} \right\} \tag{19}$$

where $\mathcal{K} = \frac{1}{1-\gamma}(\ln |\mathcal{S}| + \overline{Q}) \geq \max_W \{U(W)\}$. So, (17) can be bounded as

$$(1-\gamma) \mathbb{E}_{(W_t \sim \Pi^e, \Pi^\alpha)} \left[ \sum_{t=1} \gamma^t \big( V_Q^e(W_t) + \gamma U(W_{t+1}) - U(W_t) \big) \right]$$

$$\leq \frac{\gamma^2 (\ln |\mathcal{S}| + \overline{Q})}{(1-\gamma)} \max_S \left\{ \sqrt{2 \ln 2 \mathbf{KL}(\Pi^\alpha(\cdot|S)||\widetilde{\Pi}^\alpha(\cdot|S))} \right\} \tag{20}$$

Moreover,

$$(1-\gamma) \left( \mathbb{E}_{(W_1) \sim (P^0, \Pi^\alpha)}[U(W_1)] - \mathbb{E}_{W_1 \sim (P^0, \widetilde{\Pi}^\alpha)}[U(W_1)] \right)$$

$$\leq (1-\gamma) \max_{S, A^\alpha} \left( (\Pi^\alpha(A^\alpha|S))^2 - (\widetilde{\Pi}^\alpha(A^\alpha|S))^2 \right) \mathcal{K}$$

$$\leq 2(1-\gamma) \mathcal{K} ||\Pi^\alpha(\cdot|S_t) - \widetilde{\Pi}^\alpha(\cdot|S_t)||_\infty$$

$$\leq 2(\ln |\mathcal{S}| + \overline{Q}) \max_S \left\{ \sqrt{2 \ln 2 \mathbf{KL}(\Pi^\alpha(\cdot|S)||\widetilde{\Pi}^\alpha(\cdot|S))} \right\} \tag{21}$$

Combine (17) and (20) and (21), we bound the second term of (12) as

$$\left| \mathbb{E}_{(W, S', W') \sim \rho^{e, \alpha}} [\gamma V_Q^e(W')] - \mathbb{E}_{(W, S', W') \sim \rho^{e, \widetilde{\alpha}}} [\gamma V_Q^e(W')] \right|$$

$$= \left( \frac{\gamma^2}{(1-\gamma)} + 2 \right) (\ln |\mathcal{S}| + \overline{Q}) \max_S \left\{ \sqrt{2 \ln 2 \mathbf{KL}(\Pi^\alpha(\cdot|S)||\widetilde{\Pi}^\alpha(\cdot|S))} \right\} \tag{22}$$

The last term of (12) can be bounded simply as as

$$(1-\gamma) \left| \mathbb{E}_{W_0 \sim P^0, \Pi^\alpha} [V_Q^e(W^0)] - \mathbb{E}_{W_0 \sim P^0, \Pi^\alpha} [V_Q^e(W^0)]) \right|$$

$$\leq (1-\gamma) \max_W \{V_Q^e(W)\} \max_S \left\{ \sqrt{2 \ln 2 \mathbf{KL}(\Pi^\alpha(\cdot|S)||\widetilde{\Pi}^\alpha(\cdot|S))} \right\}$$

$$\leq (1-\gamma)(\ln |\mathcal{S}| + \overline{Q}) \max_S \left\{ \sqrt{2 \ln 2 \mathbf{KL}(\Pi^\alpha(\cdot|S)||\widetilde{\Pi}^\alpha(\cdot|S))} \right\} \tag{23}$$

Combine (16), (22) and (23) we get

$$\left| \Phi(\Pi^\alpha|Q^e) - \Phi(\widetilde{\Pi}^\alpha|Q^e) \right|$$

$$\leq \left( \frac{\gamma \overline{Q}}{(1-\gamma)^2} + \frac{\gamma^2 (\ln |\mathcal{S}| + \overline{Q})}{(1-\gamma)} + (3-\gamma)(\ln |\mathcal{S}| + \overline{Q}) \right) \max_S \left\{ \sqrt{2 \ln 2 \mathbf{KL}(\Pi^\alpha(\cdot|S)||\widetilde{\Pi}^\alpha(\cdot|S))} \right\}$$

$$= \left( \left( \frac{\gamma}{(1-\gamma)^2} + \frac{\gamma^2}{1-\gamma} + (3-\gamma) \right) \overline{Q} + \left( \frac{\gamma^2}{(1-\gamma)} + 3 - \gamma \right) \ln |\mathcal{S}| \right) \max_S \left\{ \sqrt{2 \ln 2 \mathbf{KL}(\Pi^\alpha(\cdot|S)||\widetilde{\Pi}^\alpha(\cdot|S))} \right\}$$

which completes the proof. $\qquad \square$

# B  Continuous Action Space

We extend the results in Section 4.2 to provide bounds for the case of continuous action space. In this scenario, an actor-critic method is used since the policy $\Pi^Q$ cannot be computed exactly as in Prop. 2. In this case, we can use an explicit policy $\Pi$ to approximate $\Pi^Q$ instead. We then iteratively update $Q^e$ and $\Pi$ alternatively using the loss function in Prop. 5. In particular, for a fixed $Q^e$, a soft actor update $\max_\Pi \mathbb{E}_{S' \sim \Pi(\cdot|W)}[Q^e(W, S') - \ln \Pi(S'|W)]$ will bring the imitation policy $\Pi$ closer to $\Pi^Q$.

Let $\Phi(\Pi^\alpha|Q^e, \Pi)$ be the objective of IL in Eq. (8), written as a function of the allies' joint policy $\Pi^\alpha$. The following proposition establishes a bound for the variation of $|\Phi(\Pi^\alpha|Q^e, \Pi) - \Phi(\widetilde{\Pi}^\alpha|Q^e, \Pi)|$ as function of $\mathrm{KL}(\Pi^\alpha||\widetilde{\Pi}^\alpha)$, for any pair of allies' joint policies $(\Pi^\alpha, \widetilde{\Pi}^\alpha)$.

**Proposition 7** (Continuous state space $\mathcal{S}$)**.** *Given two allies' joint policies $\Pi^\alpha$ and $\widetilde{\Pi}^\alpha$ such that for every state $S \in \mathcal{S}$, $\mathrm{KL}(\Pi^\alpha(\cdot|S)||\widetilde{\Pi}^\alpha(\cdot|S)) \leq \epsilon$, the following inequality holds:*

$$\left|\Phi(\Pi^\alpha|Q^e, \Pi) - \Phi(\widetilde{\Pi}^\alpha|Q^e, \Pi)\right| \leq \left(\alpha\overline{Q} - \beta H\right)\sqrt{2\ln 2\epsilon}$$

*where $H = \inf_W \sum_{S'} \Pi(S'|W) \ln \Pi(S'|W)$, is a lower bound of the entropy of the actor policy $\Pi$.*

*Proof.* We can follow the same arguments as in the proof of Proposition 3 above, with the only difference being when we bound $V_Q^e(W)$. Here, $V_Q^e(W)$ is replaced by $V_\Pi^e(W)$ and can be bounded as

$$V_\Pi^e(W) = \sum_{S'} \Pi^\alpha(S'|W)\left[Q^e(W, S') - \log(\Pi(S'|W))\right]$$

$$\leq \inf_{W,S'}\{Q^e(W, S')\} - \inf_W\left\{\sum_{S'} \Pi(S'|W)\log(\Pi(S'|W))\right\}$$

$$\leq \overline{Q} - H \tag{24}$$

We now see that the entropy term $\sum_{S'} \Pi(S'|W) \log(\Pi(S'|W))$ is minimized when $\Pi(S'|W) = 1/|S|$, where $H = \inf_W\{H(W)\}$, and $H(W)$ is the entropy of $\Pi(\cdot|W)$, i.e., $\sum_{S'} \Pi(S'|W) \ln \Pi(S'|W)$. Therefore, the overall bound can be established by replacing the term $\ln|\mathcal{S}|$ in the discrete case by $-H$. We then obtain

$$\left|\Phi(\Pi^\alpha|Q^e, \Pi) - \Phi(\widetilde{\Pi}^\alpha|Q^e, \Pi)\right| \leq \left(\alpha\overline{Q} - \beta H\right)\max_S\left\{\sqrt{2\ln 2\mathrm{KL}(\Pi^\alpha(\cdot|S)||\widetilde{\Pi}^\alpha(\cdot|S))}\right\}$$

$\square$

Prop. 3 & 7 allow us to establish an upper bound for the IL when the allies' joint policy changes.

**Corollary 8.** *Given two allies' policies $\Pi^\alpha$ and $\widetilde{\Pi}^\alpha$ with $\mathrm{KL}(\Pi^\alpha(\cdot|S)||\widetilde{\Pi}^\alpha(\cdot|S) \leq \epsilon, \forall S \in \mathcal{S}$, then:*

$$\left|\max_{Q^e}\min_\Pi\left\{\Phi(\Pi^\alpha|Q^e, \Pi)\right\} - \max_{Q^e}\min_\Pi\left\{\Phi(\widetilde{\Pi}^\alpha|Q^e, \Pi)\right\}\right| \leq \mathcal{O}(\sqrt{\epsilon}) \qquad \textit{(continuous)}$$

*Proof.* To simplify the notation, we first prove the following result:

**Lemma 9.** *Given $\epsilon > 0$, and two functions $f(x), g(x)$ such that $|f(x) - g(x)| \leq \epsilon$ for any $x \in \mathcal{X}$ ($\mathcal{X}$ is the feasible set of $x$). The following hold true*

$$|\max_x f(x) - \max_x g(x)| \leq \epsilon$$

$$|\min_x f(x) - \min_x g(x)| \leq \epsilon$$

To prove the above lemma, let $x^f, x^g$ be optimal solutions to $\max_x f(x)$ and $\min_x g(x)$, respectively. We consider 2 cases

(i) If $\max_x f(x) \geq \min_x g(x)$, then we have

$$|\max_x f(x) - \min_x g(x)| = f(x^f) - g(x^g) \leq f(x^f) - g(x^f) \overset{(a)}{\leq} \epsilon$$

where $(a)$ is because $|f(x) - g(x)| \leq \epsilon$ for any $x \in \mathcal{X}$

(ii) If $\max_x f(x) \le \min_x g(x)$, then we have

$$|\max_x f(x) - \min_x g(x)| = g(x^g) - f(x^f) \le g(x^g) - f(x^g) \le \epsilon$$

Thus $|\max_x f(x) - \max_x g(x)| \le \epsilon$. The inequality $|\min_x f(x) - \min_x g(x)| \le \epsilon$ can be validated in the same way.

We now get back to the main proof. Since $\left|\Phi(\Pi^\alpha|Q^e) - \Phi(\widetilde{\Pi}^\alpha|Q^e)\right| \le \mathcal{O}(\sqrt{\epsilon})$ (Proposition 3), Lemma 9 above implies that

$$\left|\max_{Q^e}\{\Phi(\Pi^\alpha|Q^e)\} - \max_{Q^e}\{\Phi(\widetilde{\Pi}^\alpha|Q^e)\}\right| \le \mathcal{O}(\sqrt{\epsilon})$$

Moreover, from Proposition 7, applying Lemma 9 twice, we have the following

$$\left|\min_\Pi \left\{\Phi(\Pi^\alpha|Q^e, \Pi)\right\} - \min_\Pi \left\{\Phi(\widetilde{\Pi}^\alpha|Q^e, \Pi)\right\}\right| \le \mathcal{O}(\sqrt{\epsilon})$$

and

$$\left|\max_{Q^e} \min_\Pi \left\{\Phi(\Pi^\alpha|Q^e, \Pi)\right\} - \max_{Q^e} \min_\Pi \left\{\Phi(\widetilde{\Pi}^\alpha|Q^e, \Pi)\right\}\right| \le \mathcal{O}(\sqrt{\epsilon})$$

which completes the proof. $\qquad\square$

## C  Other Experimental Settings and Results

### C.1  MAPPO with Supervised Learning to Predict Opponent's Next States

In this approach, instead of using IQ-learn to predict opponent's next states, we employ a simple supervised learning approach. Specifically, we create a neural network of parameters $\delta$: $M_\delta^i(o_i^\alpha, a_i^\alpha)$, taking inputs as an observation and an action of allied agent $i$, and predict the next states of enemy agents in the observable area of agent $i$. The loss function can be defined as to minimize the MSE between the actual next enemy states and predicted ones, as follows:

$$\min\left\{J_{Sup}(\delta) = \sum_{(\widetilde{S}_i^{e,\text{next}}, \widetilde{o}_i^\alpha, \widetilde{a}_i^\alpha, \forall i) \sim \texttt{buffer}} \sqrt{\sum_i \left(\widetilde{S}_i^{e,\text{next}} - M_\delta^i(\widetilde{o}_i^\alpha, \widetilde{a}_i^\alpha)\right)^2}\right\}$$

### C.2  IMAX-PPO with GAIL

To predict enemies' next states, the GAIL algorithm by [12] can be adapted in a similar way as our main IMAX-PPO algorithm. Building upon our *new MDP* framework, which is necessary to handle the unobservable-action issue, the aim is also to learn a policy $\Pi(S'|W)$ by creating a discriminator $\mathcal{D}(W, S')$ that distinguishes between generated transitions $(W, S')$ and those collected from interacting with the actual enemy agents. The GAIL loss function can be formulated as

$$\min_\Pi \max_D \left\{J_{\text{GAIL}}(\Pi, D) = \mathbb{E}_{(S',Q)\sim\rho^{\Pi,\alpha}}\left[\log D(S', W)\right] + \mathbb{E}_{\rho_{e,\alpha}}\left[\log(1 - D(W, S'))\right] - \lambda H(\Pi)\right\},$$

where $H(\Pi)$ is the entropy of $\Pi$. In practice with partial observations, we can model $\mathcal{D}$ and $\Pi$ as neutral networks $\mathcal{D}_w(S_i^{e,\text{next}}, o_i^\alpha, a_i^\alpha)$ and $\Pi_\theta(S_i^{e,\text{next}}, o_i^\alpha, a_i^\alpha)$, where $w$, $\theta$ are trainable parameters. The objective can be formulated as follows:

$$\min_\theta \max_w \left\{J_{\text{GAIL}}(\theta, w) = \mathbb{E}_{(S_i^{e,\text{next}}, o_i^\alpha, a_i^\alpha)\sim\rho^{\Pi,\alpha}}\left[\log D(S_i^{e,\text{next}}, o_i^\alpha, a_i^\alpha)\right]\right.$$

$$\left. + \mathbb{E}_{(S_i^{e,\text{next}}, o_i^\alpha, a_i^\alpha)\sim\texttt{buffer}}\left[\log(1 - D(S_i^{e,\text{next}}, o_i^\alpha, a_i^\alpha))\right] - \lambda H(\Pi_\theta)\right\}.$$

### C.3  Experimental Settings

For a fair comparison between our proposed algorithm and existing methods, we use the same model architecture and hyperparameters as shown in Tables 2 and 3 respectively. We also use

|  | SMACv2 | GRF | Miner |
|---|---|---|---|
| Runner | | Parallel | |
| Workers | | 8 | 32 |
| Total steps | 10e6 | 0.5e6 | 2e6 |
| Mini steps | | 1024 | |
| Evaluation | | 32 | |
| Mini batch | | 1 | |
| Device | | Cuda | |
| Mini epochs | | 5 | |
| Learning rate | | 5e-4 | 2e-4 |
| Epsilon | | 1e-5 | |
| Weight decay | | 0.0 | |
| Clip range | | 0.2 | |
| Entropy coefficient | 0.01 | 0.0 | 0.01 |
| Value coefficient | | 0.5 | |
| Weight gain | | 0.01 | |
| Weight initializer | | Orthogonal | |
| Max gradient norm | | 10.0 | |
| Gamma (GAE) | | 0.99 | |
| Lambda (GAE) | | 0.95 | |

Table 2: Hyperparameters

Value Normalization method and Recurrent Neural Network mechanism proposed by MAPPO [34], which used as an improvement method to speed up the training process. All sub-tasks (SMACv2, GRF, Miner) are trained concurrently in a GPU-accelerated HPC (High Performance Computing). Therefore, running times reported might not be accurate. In average, each sub-task requires 4-7 days of training depending on its difficulty. As a result, due to the limitation of our computing resources, we do not test our method's performance with other settings such as disabling Value Normalization, using MLP instead of Recurrent Neural Network, tuning learning rate, tuning clip range, etc.

Regarding GRF tasks, there are 11 academy scenarios (depicted at `https://github.com/google-research/football/blob/master/gfootball/doc/scenarios.md`). Although testing in all these sub-tasks is interesting, but we only evaluate on top three importance scenarios: *academy_3_vs_1_with_keeper*, *academy_counterattack_easy*, and *academy_counterattack_hard*. All are the hardest sub-tasks with the highest number of agents except the *almost* full football scenarios *academy_single_goal_versus_lazy*. We report the win-rate curves of our IMAX for the three tasks in Figure 4. We do not show the performance curves of the other baselines methods (QMIX, QPLEX, and MAPPO) as they are not available in their papers, noting that the final win-rates of all the methods considered are already reported in the main paper. Finally, Figure 5 shows the win-rate curves on the Gold Miner tasks.

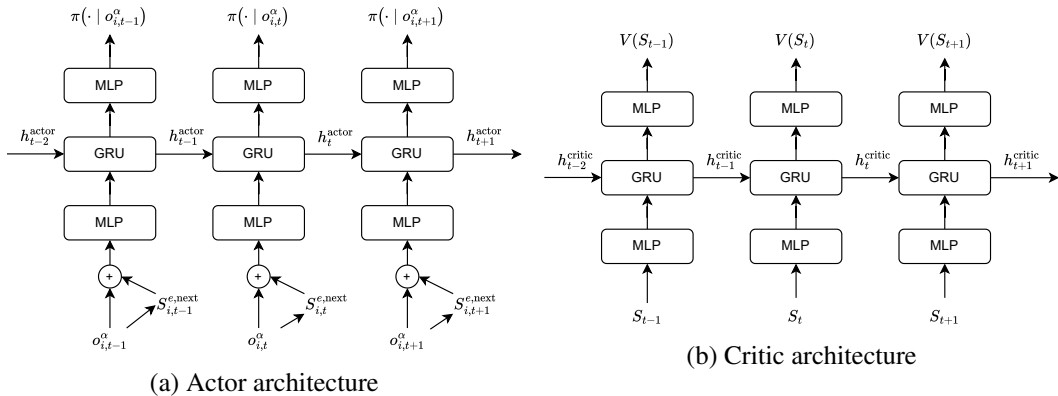

(a) Actor architecture

(b) Critic architecture

Figure 3: PPO model architecture

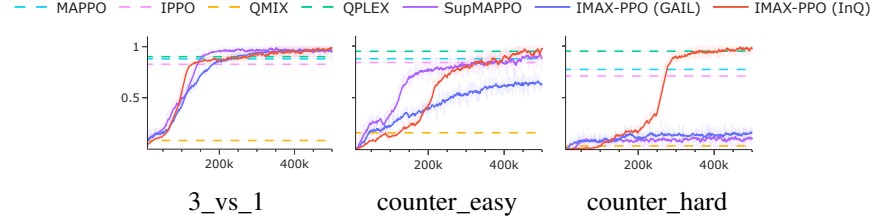

3_vs_1  counter_easy  counter_hard

Figure 4: Win-rate curves on GRF environment.

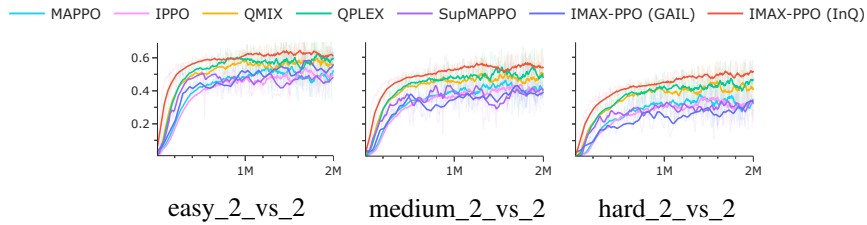

easy_2_vs_2  medium_2_vs_2  hard_2_vs_2

Figure 5: Win-rate curves on Gold Miner environment.

## D  Ablation Study

In this section, we present an ablation study to evaluate the effectiveness of our IMAX framework when integrated with traditional multi-agent reinforcement learning algorithms. Specifically, we introduce a variant called QMIX-IMAX, which incorporates our IMAX framework into the conventional QMIX algorithm. We conducted experiments using the SMACv2 benchmark, focusing on three different factions: Protoss, Terran, and Zerg. For each faction, we tested two team configurations: 5_vs_5 and 10_vs_10.

The performance of QMIX-IMAX was compared against the original, non-IMAX versions of popular algorithms such as MAPPO and QMIX. The results of these experiments are summarized in Table 3, which illustrates the winning rates achieved by each algorithm across the different scenarios. Additionally, Figure 6 displays the learning curves, providing a visual representation of the performance dynamics over the training period.

Our findings indicate that the QMIX-IMAX variant consistently outperforms the traditional QMIX algorithm across all tested scenarios. However, it does not achieve the same level of performance as our MAPPO-IMAX variant. This suggests that while the IMAX framework enhances the capabilities of QMIX, the integration with MAPPO yields superior results. These observations underscore the potential of the IMAX framework to improve baseline algorithms, offering a promising direction for future research in multi-agent reinforcement learning.

| Task | | MAPPO | QMIX | MAPPO-IMAX (ours) | QMIX-IMAX (ours) |
|---|---|---|---|---|---|
| protoss | 5_vs_5 | 21.66±9.05 | 33.84±4.90 | **48.22±7.87** | 40.75±2.99 |
| | 10_vs_10 | 9.11±2.04 | 18.64±5.97 | **46.03±5.93** | 18.65±5.03 |
| terran | 5_vs_5 | 23.95±4.89 | 41.48±5.64 | **50.56±4.84** | 48.44±4.15 |
| | 10_vs_10 | 13.53±2.83 | 34.12±3.22 | **45.16±2.87** | 38.75±5.93 |
| zerg | 5_vs_5 | 13.74±6.10 | 30.37±4.24 | **36.39±3.96** | 29.50±2.03 |
| | 10_vs_10 | 6.06±1.76 | 28.13±7.76 | **41.53±6.53** | 25.52±3.81 |

Table 3: Wining rate (percentage) of two baseline methods MAPPO and QMIX and our improvement algorithm on SMACv2 tasks.

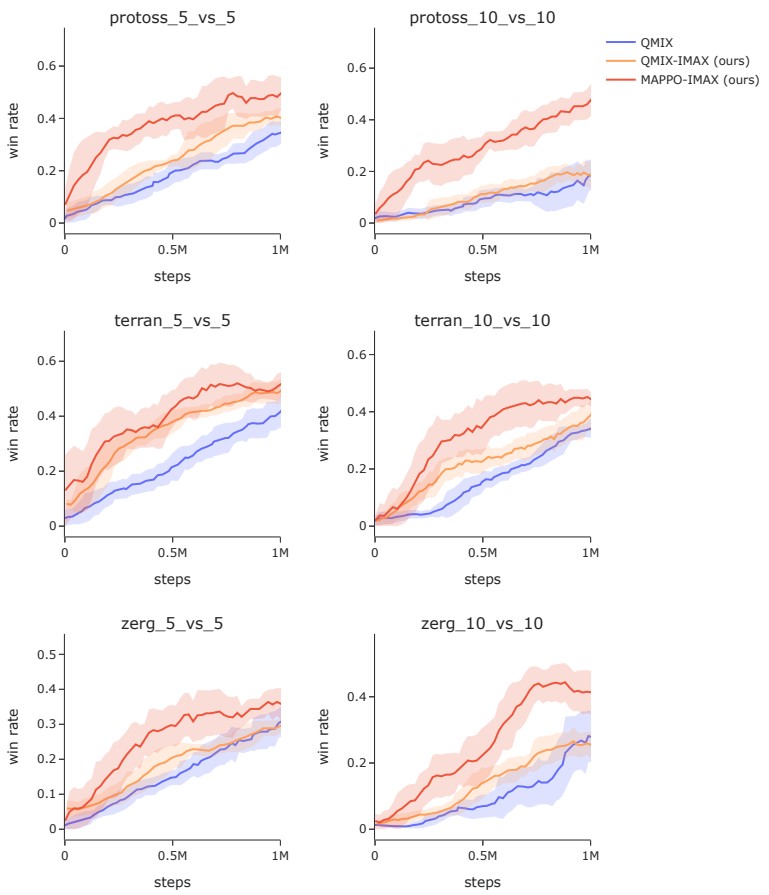

Figure 6: Learning curves with different methods on SMACv2.

