# OpenReview forum: "Mimicking To Dominate: Imitation Learning Strategies for Success in Multiagent Games"
_NeurIPS.cc/2024/Conference — NeurIPS 2024 poster_

### Official Review · Reviewer_ZqFd · 2024-07-17

**Soundness:** 3
**Presentation:** 3
**Contribution:** 2
**Rating:** 6
**Confidence:** 3

**Summary:**

This paper addresses the issue of training instability and slow convergence in MARL caused by the changing strategies of other agents. It proposes reducing the uncertainty faced during training by imitating the opponents' strategies. To address the challenge that opponents' actions are usually unobservable, the paper further proposes predicting the opponents' next state, providing corresponding derivation and analysis. Finally, the authors tested their approach in multiple experimental environments, achieving significant improvements in final rewards.

**Strengths:**

The paper is well-organized, with clear motivation and straightforward method introduction, making it easy to follow. The adaptation of IQ-Learn is easy to understand, and the derivation process is quite clear and technically sound.

The experiments included SOTA MARL algorithms and compared results under different imitation learning frameworks. In the majority of tasks across three different environments, the proposed IMAX-PPO algorithm achieved the highest expected rewards, demonstrating the effectiveness of the method.

**Weaknesses:**

I have not found any obvious flaws. My concern lies in the experiments. I noticed that the training curves in the SMACv2 and Gold Miner environments are very unstable, i.e., the performance flucuates. Is this due to adversarial imitation learning? The stability of the algorithm needs further validation. Additionally, the default settings have fixed enemy strategies, making the learning easier. If the enemies are also learning agents, the instability may be exaggerated.

**Questions:**

1. The performance of many methods in the experiments fluctuate. How many seeds were used in the experiments? Previous experiments on SMACv2 usually uses more than 5 random seeds.
2. Are the strategies of the enemies fixed in the environment?

**Limitations:**

The training process may be unstable.

---

> ### Author Rebuttal · Authors · 2024-08-07
>
> We thank the reviewer for reading our paper and providing positive feedback.
>
> > The performance of many methods in the experiments fluctuate. How many seeds were used in the experiments? Previous experiments on SMACv2 usually uses more than 5 random seeds.
>
> The main reason the win-rate curves are not very stable and fluctuate at some points is that MARL tasks are highly complex, particularly SMAC_v2, which is considered one of the most challenging tasks in MARL. This is why all the curves of all the methods are not as stable as those typically seen in single-agent training. Additionally, the inclusion of more components in our method (such as the IL component) can also contribute to instability. Despite this instability, our method consistently converges to better policies compared to other baselines.
>
> Regarding the number of seeds, we followed the standard setting in prior MARL papers to train and test our algorithm. We used 5 seeds for training and 32 seeds to evaluate the win rate for each given policy. We will clarify this in the paper.
>
> > Are the strategies of the enemies fixed in the environment?
>
> Yes, all the enemy policies are fixed in the environment. This is a standard setting employed in other state-of-the-art MARL algorithms such as MAPPO and QMIX. Allowing enemies to learn and adapt their policies during training would be an interesting configuration, but it would require significant effort, as the current environmental setup (such as SMACv2) does not support this. Additionally, from an algorithmic design perspective, a new MARL algorithm would be needed to handle this situation. We plan to explore this in future work.
>
> *We hope the above responses are clear. If you have any other questions or concerns, we are happy to address and clarify them.*

---

> > ### Comment · Reviewer_ZqFd · 2024-08-09
> >
> > Thanks for the reply. My concern on the stability issue has not been fully dispelled. Although I agree with the authors' response that the inclusion of more components may result in instability, I wonder whether IMAX-PPO framework needs carefully hyper-parameters tuning or the help of complex training tricks.

---

> > > ### Author Response · Authors · 2024-08-09
> > >
> > > We thank the reviewer for the prompt reply.
> > >
> > > > I wonder whether IMAX-PPO framework needs carefully hyper-parameters tuning or the help of complex training tricks.
> > >
> > > Thank you for the question. We would like to clarify that we did not conduct much hyper-parameter tuning for the IMAX-PPO. Instead, we adopted some default parameter settings from the MAPPO paper. For the IL component, we also utilized the  default parameters from the IQ-learn paper. We would also like to note that the inclusion of the IL component can make training more computationally expensive and more unstable compared to other baselines. However, we believe that the gains in the recovered policy are worth these trade-offs.
> > >
> > > *We hope the above answers your question. If you have any further questions, we are happy to clarify them.*

---

### Official Review · Reviewer_4odM · 2024-07-19

**Soundness:** 4
**Presentation:** 4
**Contribution:** 4
**Rating:** 8
**Confidence:** 3

**Summary:**

This paper presents a new framework of multi-agent reinforcement learning (MARL) by modeling opponents’ behaviors through imitation learning.

**Strengths:**

The motivation and the method is well described and the performance is tested in extensive experiments with challenging tasks against SOTA methods.

**Weaknesses:**

It is assumed that each agent performs just based on the present observation o_i, but coordinated behaviors often require memory coding the group strategy or opponents’ game plan.

**Questions:**

If I understand correctly, an important feature of the proposed framework is that all ally agents jointly learns a single joint model of the enemies. On the other hand, the SupMAPPO agents learn the enemies’ next states individually. What if the same information sharing is performed for supervised prediction of enemies’ next state?

**Limitations:**

It is assumed that opponents don’t learn.

---

> ### Author Rebuttal · Authors · 2024-08-07
>
> We thank the reviewer for reading our paper and for the positive  feedback.
>
> > If I understand correctly, an important feature of the proposed framework is that all ally agents jointly learns a single joint model of the enemies. On the other hand, the SupMAPPO agents learn the enemies’ next states individually. What if the same information sharing is performed for supervised prediction of enemies’ next state?
>
> In response to your question, we would like to clarify that in both of our main algorithms, IMAXPPO and SupMAPPO, we use a common joint model for all the enemies. Although each enemy agent provides different inputs, leading to different outputs, this common joint model helps reduce the size of the learning model and is more efficient than learning individual models for each enemy agent.
>
> >It is assumed that opponents don’t learn.
>
> Yes, all the enemy policies are fixed in the environment. This is a standard setting employed in other state-of-the-art MARL algorithms such as MAPPO and QMIX. Allowing enemies to learn and adapt their policies during training would be an interesting configuration, but it would require significant effort, as the current environmental setup (such as SMACv2) does not support this. Additionally, from an algorithmic design perspective, a new MARL algorithm would be needed to handle this situation. We plan to explore this in future work.
>
> *We hope the above responses are clear. If you have any other questions or concerns, we are happy to address and clarify them.*

---

### Official Review · Reviewer_bLoo · 2024-07-29

**Soundness:** 2
**Presentation:** 1
**Contribution:** 3
**Rating:** 6
**Confidence:** 3

**Summary:**

The paper studies cooperative-competive MARL. It utilizes imitation learning to comprehend and anticipate the next actions of the opponent agents (enemies), aiming to mitigate uncertainties of the controlled agents (allies) with respect to the game dynamics.

**Strengths:**

- The paper studies a very interesting problem, central to the MARL community.
- The paper proposes a novel and interesting method, combining imitation learning techniques and opponent modelling for enhancing the agents' individual policies.
- The paper provides many experiments on three benchmarks (smacv2, grf, gold miner)
- The proposed method significantly improves performance in many tasks over the most important baselines: the backbone MAPPO and the supervised baseline, SUP-MAPPO.
- The authors provide interesting theoretical analysis of the proposed framework.

**Weaknesses:**

- Related work needs improvement. In opponent modelling, the authors claim that: "All the aforementioned related works require having access to opponent’ observations and actions during training and/or execution". This is not the case in most opponent/agent modelling works (e.g., see Papoudakis et al.), as they model the other allies, not the enemies (which is allowed under the CTDE schema). I believe that the proposed method belongs to the category of opponent/agent modelling in MARL. Moreover, some important references are missing, see [1], [2], [3].
- The presentation needs improvement, heavy notation in many parts. The authors should remind more often what some quantities represent. Furthermore, more intuition of the proposed framework and the method is needed in sections 4 and 5 and the related work. In a nutshell, why can IL solve the agent modelling problem, why is it important and how? Also, how does the goal of the method (i.e., predicting the next states of the enemies) is connected to the IL objective of section 4.1?
- Ablation study is missing. Also, it would be interesting if the authors provide experiments of the proposed method on top of other MARL algorithms as well.
- The framework can be impractical as it is now, as it may need a lot of hardcoding to be implemented to any environment, since it leverages enemy state information within the individual (allies) agents' observations (e.g., information regarding the neighborhoods). In other words, one may need to be able to decompose manually the agents' observations into different parts, some of them are related to enemies.

[1] Papoudakis, Georgios, Filippos Christianos, and Stefano Albrecht. "Agent modelling under partial observability for deep reinforcement learning." Advances in Neural Information Processing Systems 34 (2021): 19210-19222.

[2] J. Sun, S. Chen, C. Zhang, Y. Ma, and J. Zhang, “Decision-making with speculative opponent 388 models,” IEEE Transactions on Neural Networks and Learning Systems, 2024.

[3] R. Raileanu, E. Denton, A. Szlam, and R. Fergus, “Modeling others using oneself in multi-agent reinforcement learning,” in International conference on machine learning. PMLR, 2018, pp. 4257–4266.

**Questions:**

The questions have been intergrated into the weaknesses section.

**Limitations:**

Limitations of the work have been discussed in the weaknesses section. The authors also highlight some limitations in the conclusion.

---

> ### Author Rebuttal · Authors · 2024-08-07
>
> We thank the reviewer for carefully reading our paper and providing us with constructive feedback.
> > Related work needs improvement. In opponent modelling, the authors claim that: "All the aforementioned related works require having access to opponent’ observations and actions during training and/or execution". This is not the case in most opponent/agent modelling works ...
>
> Thank you for your feedback! Yes, our work falls under the category of opponent modeling in MARL. We will revise the section to clearly differentiate between opponent-agent modeling and allied-agent modeling. In our competitive MARL setting in particular, access to opponents’ observations and actions during both training and execution is not allowed, thus making our opponent-prediction significantly more challenging.
>
> We will incorporate discussions on suggested references. Specifically, Reference [1] relies on having access to the observations and actions of the modeled agents during training to develop accurate representations. In contrast, our setting does not provide such inputs, making our task more challenging. Reference [2] involves training the opponent model in conjunction with the learning agents’ policies to mainly maximize the agents’ expected return, without reasoning about the opponents’ long-term goals and plans. On the other hand, our approach takes a more sophisticated perspective by utilizing information about opponents derived from our agents’ local observations. We employ imitation learning to uncover and understand the long-term, complex policies of the opponents, providing a deeper insight into their behavior. Finally, reference [3] assumes that all agents have full visibility of the environment, while in our scenario, each agent is limited to local observations.
>
> > The presentation needs improvement, heavy notation in many parts. The authors should remind more often what some quantities represent. Furthermore, more intuition of the proposed framework and the method is needed in sections 4 and 5 and the related work. In a nutshell, why can IL solve the agent modeling problem, why is it important and how? Also, how does the goal of the method (i.e., predicting the next states of the enemies) be connected to the IL objective of section 4.1?
>
> We will add more intuitive explanations and revise the notations as suggested. Our rationale for employing imitation learning (IL) to predict opponents' behavior stems from IL’s proven effectiveness in understanding and replicating the long-term goals and complex policies of agents based on historical trajectory observations. Especially, in our MARL setting, IL has demonstrated its significant advantages in predicting the opponents’ behavior solely based on partial observations about these opponents, resulting in a remarkable performance of our MARL algorithm, IMAX-PPO, substantial outperforming other SOTA algorithm in complex MARL environments such as SMACv2 and Google Research Football.
>
> The objective in Section 4.1 is to learn an IL policy $\Pi(S'|S, A^\alpha)$ from demonstrations by treating next-state prediction as an IL problem. This policy allows for the prediction of the next state of the enemies given the current state $S$ and ally action taken in the previous round. This information is then used in our IMAX-MAPPO algorithm to support the MARL.
>
> We hope the above responses are clear. If you have any other questions, we are happy to address and clarify them.
>
> > Ablation study is missing. Also, it would be interesting if the authors provide experiments of the proposed method on top of other MARL algorithms as well.
>
> In the current paper, we have included an ablation study comparing the performance of our IMAX-MAPPO algorithm with other IL-based and supervised-learning-based approaches (Sup-MAPPO, GAIL-MAPPO). Additionally, in response to your question about incorporating our IL-based approach with other MARL algorithms, we have provided additional experiments for QMIX-IMAX, as reported in the attached 1-page PDF. The results generally show that our IL-based framework can improve QMIX, but this combination is generally less effective than our main algorithm, IMAX-MAPPO. We will include such results in the paper.
>
> > The framework can be impractical as it is now, as it may need a lot of hardcoding to be implemented to any environment, since it leverages enemy state information within the individual (allies) agents' observations (e.g., information regarding the neighborhoods). In other words, one may need to be able to manually decompose the agents' observations into different parts, some of them related to enemies.
>
> Since we have more components to learn (imitation learning to mimic the opponent's policy), we need an additional step to decompose the agents’ observations into different parts. This ensures that the appropriate information is used for imitation learning or main policy learning. So far, this decomposition step has not required significant effort and can be done with just a few lines of code. However, it would be ideal to have an automated algorithm that can perform the decomposition automatically. We will investigate this in future work.
>
> *We hope the above responses are clear. If you have any other questions, we are happy to address and clarify them.*

---

> > ### Comment · Reviewer_bLoo · 2024-08-09
> >
> > I would like to thank the authors for addressing my concerns and questions. I have increased my rating score.

---

> > > ### Author Response · Authors · 2024-08-10
> > >
> > > We would like to thank the reviewer for reading our responses and for maintaining a positive view of our paper!

---

### Author Rebuttal · Authors · 2024-08-07

We thank the reviewers for carefully reading our paper and providing constructive feedback, which we have made efforts to address. Please find a summary of our responses below.

**Reviewer bLoo** mentioned some related work in opponent/agent modeling. In response, we have explicitly provided a discussion on the differences between our IL-based method and those in the opponent/agent modeling literature. The reviewer also raised questions about why IL is useful for solving the agent modeling problem and for effective MARL, as well as the practicality of our method when decomposing the information to facilitate learning. We have provided detailed responses to these questions. Additionally, the reviewer asked about applying the IL-based framework on top of other MARL algorithms. This point is well taken, and in the attached 1-page PDF, we have provided additional experiments for QMIX+IMAX. The experiments generally show that incorporating IMAX can improve QMIX, but this combination is generally outperformed by our main algorithm (MAPPO-IMAX).

**Reviewer 4odM** asked whether SupMAPPO agents learn the enemies' next states using a joint enemy agent model. In response, we clarified that both SupMAPPO and our IMAX-MAPPO use a single joint model of the enemies. This approach is more practical than using an individual model for each agent.

**Reviewer ZqFd** questioned why our learning/evaluation curves are not very stable and how many seeds were used in the experiments. We have provided detailed responses to these questions and will add clarifications to the paper.

We thank all the reviewers for their comments and positive feedback, which we have tried to address and clarify. If you have any further questions, we are more than happy to discuss and clarify them.

---

### Decision · Program_Chairs · 2024-09-25

**Decision:**

Accept (poster)

**Comment:**

All reviewers agree this is a good paper, on a relevant topic, with substantial contributions. There is nevertheless an opportunity to improve clarity and presentation, as raised in some concrete instances by multiple reviewers, for the camera-ready.